

# Uncertainty analysis of a European high-resolution emission inventory of CO₂ and CO to support inverse modelling and network design

Ingrid Super[1], Stijn N.C. Dellaert[1], Antoon J.H. Visschedijk[1], Hugo A.C. Denier van der Gon[1]

[1]Department of Climate, Air and Sustainability, TNO, P.O. Box 80015, 3508 TA Utrecht, Netherlands

*Correspondence to*: Ingrid Super (ingrid.super@tno.nl)





**Abstract.** Quantification of greenhouse gas emissions is receiving a lot of attention, because of its relevance for climate mitigation. Quantification is often done with an inverse modelling framework, combining atmospheric transport models, prior gridded emission inventories and a network of atmospheric observations to optimize the emission inventories. An important aspect of such method is a correct quantification of the uncertainties in all aspects of the modelling framework. The uncertainties in gridded emission inventories are, however, not systematically analysed. In this work, a statistically coherent method is used to quantify the uncertainties in a high-resolution gridded emission inventory of $CO_2$ and CO for Europe. We perform a range of Monte Carlo simulations to determine the effect of uncertainties in different inventory components, including the spatial and temporal distribution, on the uncertainty in total emissions and the resulting atmospheric mixing ratios. We find that the uncertainty in the total emissions for the selected domain are 1 % for $CO_2$ and 6 % for CO. Introducing spatial disaggregation causes a significant increase in the uncertainty of up to 40 % for $CO_2$ and 70 % for CO for specific grid cells. Using gridded uncertainties specific regions can be defined that have the largest uncertainty in emissions and are thus an interesting target for inverse modelers. However, the largest sectors are usually the best-constrained ones (low relative uncertainty), so the absolute uncertainty is the best indicator for this. With this knowledge areas can be identified that are most sensitive to the largest emission uncertainties, which supports network design.





## 1 Introduction

Carbon dioxide ($CO_2$) is the most important greenhouse gas and is emitted in large quantities from human activities, especially from the burning of fossil fuels (Berner, 2003). A reliable inventory of fossil fuel $CO_2$ (FFCO$_2$) emissions is important to increase our understanding of the carbon cycle and how the global climate will develop in the future. The impact of $CO_2$ emissions is visible on a global scale and international efforts are required to mitigate climate change, but cities are the largest contributors to FFCO$_2$ emissions (about 70% (IEA, 2008)). Therefore, emissions should be studied at different spatial and temporal scales to get a full understanding of their variability and mitigation potential.

One way of describing emissions is an emission inventory, which is a structured set of emission data, distinguishing different pollutants and source categories. Often, emission inventories are based on reported country-level emission data (for example from the National Inventory Reports (NIR's)), which are spatially and temporally disaggregated (scaled-down) to a certain level using proxies (e.g. the TNO inventories (Denier van der Gon et al., 2017; Kuenen et al., 2014)). Other emission inventories are based on local energy consumption data and reported emissions aggregated to the required spatial scale (scaled-up) (e.g. Hestia (Gurney et al., 2011, 2019)) or rely on (global) statistical data and a consistent set of (non-country specific) emission factors representing different technology levels (e.g. EDGAR (http://edgar.jrc.ec.europa.eu)). Most inventories, including the one used in this study, rely on a combination of methods, using large-scale data supplemented with local data. Gridded emission inventories are essential as input for atmospheric transport models to facilitate comparison with observations of $CO_2$ concentrations, as well as in inverse modelling as a prior estimate of the emission locations and magnitude.

During the compilation of an emission inventory uncertainties are introduced at different levels (e.g. magnitude, timing or locations) and increasingly more attention is given to this topic. Parties to the United Nations Framework Convention on Climate Change (UNFCCC) report their annual emissions (disaggregated over source sectors and fuel types) in a NIR (UNFCCC, 2019), which includes an assessment of the uncertainties in the underlying data and an analysis of the uncertainties in the total emissions following IPCC (Intergovernmental Panel on Climate Change) guidelines. The simplest uncertainty analysis is based on simple equations for combining uncertainties from different sources (Tier 1 approach). A more advanced approach is a Monte Carlo simulation, which allows for non-normal uncertainty distributions (Tier 2 approach). The Tier 2 approach has been used by several countries, for example Finland (Monni et al., 2004) and Denmark (Fauster et al., 2011).

These reports provide a good first step in quantifying emission uncertainties, but the uncertainty introduced by using proxies for spatial and temporal disaggregation are not considered. These are, however, an important source of uncertainty in the gridded emission inventories (Andres et al., 2016). Inverse modelling studies are increasingly focusing on urban areas, the main source areas of FFCO$_2$ emissions, for which emission inventories with a high spatiotemporal resolution are used to better represent the variability in local emissions affecting local concentration measurements. Understanding the uncertainty at higher resolution than the country-level is thus necessary, which means that the uncertainty caused by spatiotemporal disaggregation becomes important as well. The uncertainties in emission inventories are important to understand for several reasons. First, knowledge of uncertainties helps pinpointing emission sources or areas that require more scrutiny (Monni et al., 2004; Palmer et al., 2018). Second, knowledge of uncertainties in prior emission estimates is an important part of inverse modelling frameworks, which can be used for emission verification and in support of decision-making (Andres et



al., 2014). For example, if uncertainties are not properly considered, there is a risk that the uncertainty range does not contain the actual emission value. In contrast, if uncertainties are large the prior gives little information about the actual emissions and more independent observations are needed. Third, emission uncertainties can support atmospheric observation system design, for example for inverse modelling studies. An ensemble of model runs can represent the spread in atmospheric concentration fields due to the uncertainty in emissions. Locations with a large spread in atmospheric concentrations are most sensitive to uncertainties in the emission inventory and are preferential locations for additional atmospherics measurements. To conclude, emission uncertainties are a critical part of emission verification systems and require more attention. To better understand how uncertainties in underlying data affect the overall uncertainty in gridded emissions, a family of ten emission inventories is compiled within the H2020 project $CO_2$ Human Emissions (CHE). These can be made available upon request.

In this paper we illustrate a statistically coherent method to assess the uncertainties in a high-resolution emission inventory, including uncertainties resulting from spatiotemporal disaggregation. For this purpose, we use a Monte Carlo simulation to propagate uncertainties in underlying parameters into the total uncertainty in emissions (like the Tier 2 approach). We illustrate our methodology using a new high-resolution emission inventory for a European zoom region. We illustrate the magnitude of the uncertainties in emissions and how this affects simulated concentrations. The research questions are:

1) How large are uncertainties in total inventory emissions and how does this differ per sector and country?
2) How do uncertainties in spatial proxy maps affect local measurements?
3) How important is the uncertainty in time profiles for the calculation of annual emissions from daytime (12-16h LT) emissions, which result from inverse modelling studies using only daytime observations?
4) What information can we gain from high-resolution gridded uncertainty maps by comparing different regions?

Inverse modelling studies often focus on a single species like $CO_2$, but co-emitted species are increasingly included to allow source partitioning (Boschetti et al., 2018; Zheng et al., 2019). In this study, we look into $CO_2$ and CO to illustrate our methodology, but the methodology can be applied to other (co-emitted) species.

## 2 Methodology

### 2.1 The high-resolution emission inventory

**Table 1: Characteristics of the high-resolution emission inventory TNO GHGco v1.0**

| | |
|---|---|
| **Air pollutants** | CO_ff, CO_bf, NOx |
| **Greenhouse gases** | $CO_2$_ff, $CO_2$_bf, $CH_4$ |
| **Resolution** | 1/60° longitude x 1/120° latitude (~ 1x1 km over central Europe) |
| **Period covered** | 2015 (annual emissions) |
| **Domain** | -2° W–19° E, 47° N–56°N |
| **Sector aggregation** | GNFR (A to L), with GNFR F (Road Transport) split in F1 to F4 (total 16 sectors) |
| **Countries** | Complete: DEU, NLD, BEL, LUX, CZE |
| | Partially: GBR, FRA, DNK, AUT, POL, CHE, ITA, SVK and HUN |

The basis of this study is a high-resolution emission inventory for the greenhouse gases $CO_2$ and $CH_4$ and the co-emitted tracers CO and $NO_x$ for the year 2015 (TNO GHGco v1.0, see details in Table 1). In this paper we only





use CO₂ and CO, which are divided over fossil fuel (ff) and biofuel (bf) emissions (no land use and land use
change emissions are included). The emission inventory covers a domain over Europe, including Germany,
Netherlands, Belgium, Luxembourg and the Czech Republic, and parts of Great Britain, France, Denmark, Austria
and Poland (see also Figure 1).
The emission inventory is based on the reported emissions by European countries to the UNFCCC (only
greenhouse gases) and to EMEP/CEIP (European Monitoring and Evaluation Programme/Centre on Emission
Inventories and Projections, only air pollutants). UNFCCC $CO_2$ emissions have been aggregated to ~250 different
combinations of NFR sectors (Nomenclature For Reporting) and fuel types. EMEP/CEIP CO emissions have been
split over the same NFR sector-fuel type combinations by TNO using the GAINS model (Amann et al., 2011)
and/or TNO data. In some cases, the data was gap filled, replaced or (dis)aggregated to obtain a consistent dataset.
Next, each NFR sector is linked to a high-resolution proxy map (e.g. population density for residential combustion
of fossil fuels or AIS data for shipping re-gridded to 1/60° x 1/120°), which is used to spatially disaggregate the
reported country-level emissions. Where possible, the exact location and reported emission of large point sources
is used (e.g. from the E-PRTR (European Pollutant Release and Transfer Register)). The third step is temporal
disaggregation, for which standard time profiles are used (Denier van der Gon et al., 2011), Finally, the gridded
emissions are aggregated per GNFR sector (see Table 2) for the emission inventory. The final emission maps of
CO₂ and CO are shown in Figure 1, together with two examples of a source sector map. Note that these maps do
not clearly show the large point source emissions, while these make up almost 45 % of all CO₂ emissions and 26
% of all CO emissions.

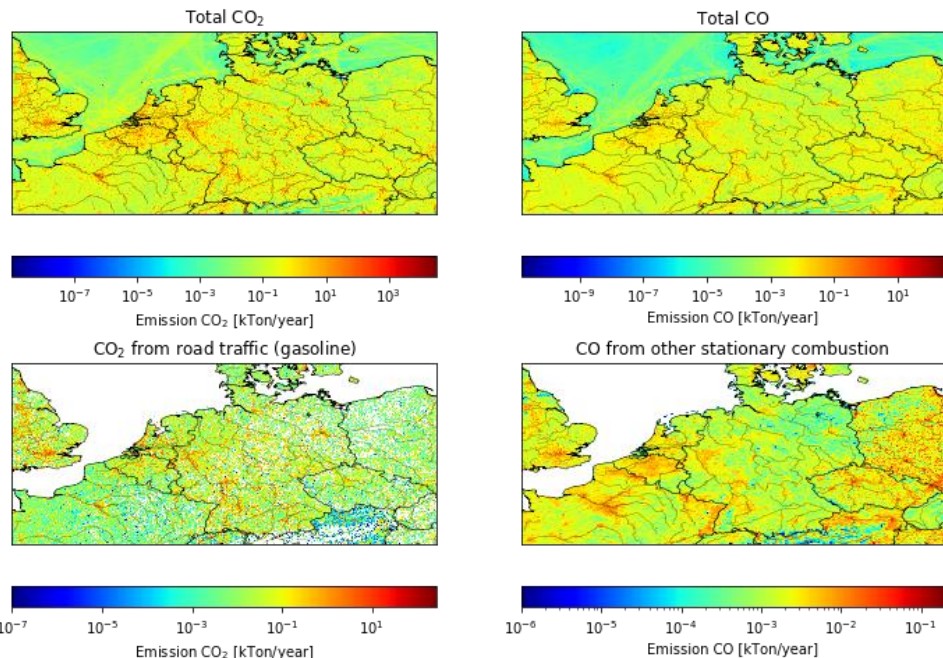


**Figure 1: Total emissions of CO₂ and CO, road traffic (gasoline) emissions of CO₂, and other stationary combustion emissions of CO for 2015 in kt yr⁻¹ (defined per grid cell).**





**2.2 Uncertainties in parameters**

The emission inventory is used as basis for an uncertainty analysis by assigning an uncertainty to each parameter underlying the UNFCCC-EMEP/CEIP emission inventories and further disaggregation thereof. Although the aggregation to GNFR sectors makes the emission inventory more comprehensible, we use the more detailed underlying data for the uncertainty analysis. The reason is that the uncertainties can vary enormously between sub-sectors and fuel types. Generally, the emission at a certain time and place is determined by four types of parameters: activity data, emission factor, spatial distribution and time profile. The activity data and emission factors are used by countries to calculate their emissions. The spatial proxy maps and time profiles are used for spatiotemporal disaggregation. All uncertainties need to be specified per NFR sector-fuel type combination that is part of the Monte Carlo simulation. In the following sections the steps taken to arrive at a covariance matrix for the Monte Carlo simulation are described. Tables with uncertainty data can be found in the Appendix.

**Table 2: Overview of GNFR sectors distinguished in the emission inventory**

| GNFR category | GNFR category name |
|---|---|
| A | A_PublicPower |
| B | B_Industry |
| C | C_OtherStationaryComb |
| D | D_Fugitives |
| E | E_Solvents |
| F | F_RoadTransport |
| G | G_Shipping |
| H | H_Aviation |
| I | I_OffRoad |
| J | J_Waste |
| K | K_AgriLivestock |
| L | L_AgriOther |
| F1 | F_RoadTransport_exhaust_gasoline |
| F2 | F_RoadTransport_exhaust_diesel |
| F3 | F_RoadTransport_exhaust_LPG_gas |
| F4 | F_RoadTransport_non-exhaust |

**2.2.1 Parameter selection**

The first step is to identify which parameters should be included in the Monte Carlo simulation. As mentioned before there are about 250 different combinations of NFR sectors and fuel types and including all of them would be a huge computational challenge. However, a selection of 112 combinations makes up most of the fossil fuel emissions (96 % for $CO_2$ and 92 % for CO) and therefore a pre-selection was made. This results in a covariance matrix of 224x224 parameters (112 sector-fuel combinations for two species). To further reduce the size of the problem, the emissions are partly aggregated before starting the Monte Carlo for the spatial proxies (mostly fuels are combined per sector, because they have the same spatial distribution). This results in a total of 59 NFR sector-spatial proxy combinations, which are put in a separate covariance matrix. The time profiles are applied to the



aggregated GNFR sectors, which make up the last covariance matrix. Note that the spatial proxies and time
profiles are the same for $CO_2$ and CO, except for the spatially explicit E-PRTR point source data.

**2.2.2 Uncertainties in reported emissions**

Country-level emissions are estimated from the multiplication of activity data and emission factors. Activity data
consist for the most part of fossil fuel consumption data available from national energy balances. Some fuel
consumptions are better known than others and uncertainties vary across sectors. An emission factor is the amount
of emission that is produced per unit of activity (e.g. amount of fuel consumed). For $CO_2$ this depends mainly on
the carbon content of the fuel. In contrast, CO emissions are extremely dependent on combustion conditions,
certain industrial processes and in-place technologies.
The NIR's for greenhouse gases (GHGs) provide a table with uncertainties in activity data and $CO_2$ emission
factors on the level of NFR sector - fuel combinations. The uncertainties reported by each country are averaged
to get one uncertainty per NRF sector-fuel combination for the entire domain. Overall, the differences between
countries are small. The uncertainties in activity data and $CO_2$ emission factors are relatively low and normally
distributed.
The CO emission factors are mostly based on basic uncertainty ranges provided in the EMEP/EEA Guidebook
(European Environment Agency, 2016) and supplemented by BAT reference documents from which reported
emission factor ranges are taken as uncertainty range (http://eippcb.jrc.ec.europa.eu/reference/). The CO emission
factor uncertainties are generally expressed by a factor, which means that the highest and lowest limit values are
either the specified factor above or below the most common value. Therefore, these uncertainties have a lognormal
distribution and are relatively large.

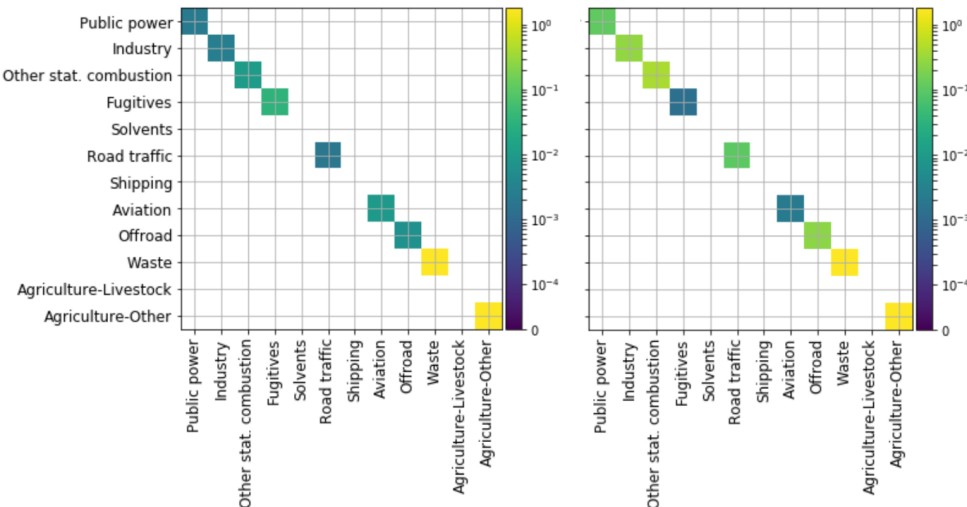


**Figure 2: Covariance matrices for total emissions of $CO_2$ (left) and CO (right) per aggregated source sector. A white space on the diagonal indicates this sector is not included in the Monte Carlo simulation.**

To estimate the overall uncertainty in the emissions per NFR sector-fuel combination, the uncertainties in the
activity data and emission factors need to be combined (shown in Figure 2 for the aggregated GNFR sectors).
When both uncertainties are of the same order and relatively small, as well as both having a normal distribution,





the overall emission uncertainty is calculated with the standard formula for error propagation for non-correlated
normally distributed variables (see Sect. 2.4). For most CO emission factors, uncertainties are much higher and
have a lognormal distribution instead of normal. In that case the uncertainty of the variable with the highest
uncertainty is assumed to be indicative for the overall uncertainty of the emission, which in general means the
uncertainty of the CO emission factor determines the overall uncertainty of the CO emission, with the distribution
remaining lognormal. The error introduced by fuel type disaggregation for CO is not considered.
Finally, for power plants and road traffic we assumed error correlations to exist between different sub-sectors per
fuel type, and between different fuel types per sub-sector for other NFR sectors. In some cases, correlations also
exist between different NFR sectors belonging to the same GNFR sector. The definition of correlations is
important, because they affect the total uncertainties. For example, if emission factors of sub-sectors are
correlated, deviations can amplify each other, leading to higher overall uncertainties. In contrast, the division of
the well-known total fuel consumption of a sector over its sub-sectors includes an uncertainty which is anti-
correlated (i.e. if too much fuel consumption is assigned to one sub-sector, too little is assigned to another). This
has little impact on the total emissions, because uncertainties only exist at lower levels.

### 2.2.3 Uncertainties in spatial proxies

The proxy maps used for spatial disaggregation can introduce a large uncertainty coming from the following
sources:
1)    The proxy is not correctly representing real-world locations of what it is supposed to represent, either
because there are cells included in which none of the intended activity takes place or cells are missing in
which the intended activity does take place (proxy quality).
2)    The proxy is not fully representative for the activity it is assumed to represent, for example if there is a non-
linear relationship between the proxy value and the emission (proxy representativeness): a grid cell with
twice the population density does not necessarily have double the amount of residential heating emissions,
because heating can be more efficient in densely populated areas and/or apartment blocks.
3)    The cell values themselves are uncertain, e.g. the population density or traffic intensity (proxy value).
We attempt to capture the second and third source of uncertainty in a single numerical indicator representing the
uncertainty at cell level (see Figure 3 for the uncertainty per aggregated GNFR sector). The overall uncertainties
are based on expert judgement and inevitably include a considerable amount of subjectivity. This type of
uncertainty is often large and has a lognormal distribution, except for proxies related to road traffic and some
proxies related to commercial/residential emissions sources. We assume no error correlations exist. The first
source of uncertainty is also considered in one of the experiments (see Sect. 2.4 for a description of this
experiment).

### 2.2.4 Uncertainties in time profiles

The time profiles currently consist of fixed monthly, daily and hourly fractions that are based on long-term average
activity data and/or socio-economic characteristics. These profiles are applied for each year and for the entire
domain, considering only time zone differences. In reality, the time profiles can differ between countries, and
from year to year. For example, residential emissions are strongly correlated with the outside temperature and





therefore show a strong seasonal cycle. However, one winter can be very cold, whereas the next can have above-
average temperatures.

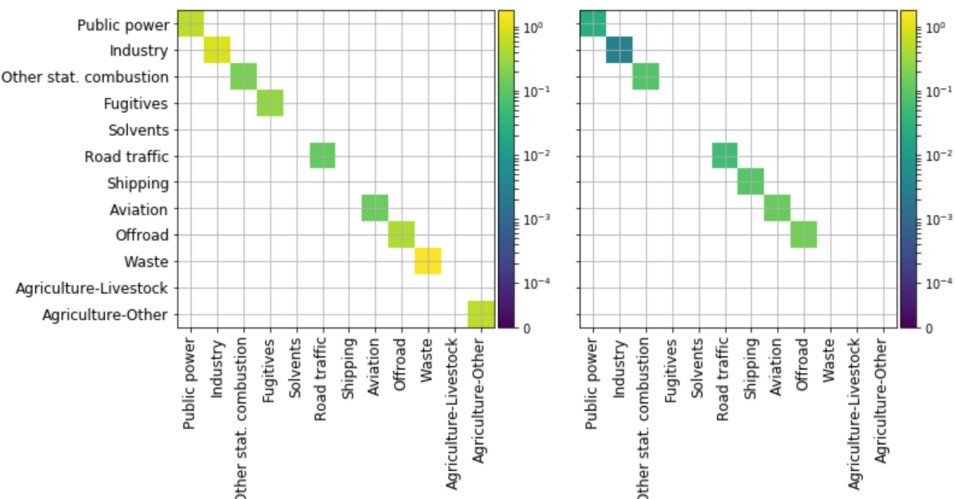

**Figure 3: Covariance matrices for spatial proxies (left) and time profiles (right) per aggregated source sector. These are the same for $CO_2$ and CO. A white space on the diagonal indicates this sector is not included in the Monte Carlo simulation.**

To quantify the uncertainty in time profiles, a range of time profiles (for a full year, hourly resolution) was created
for each source sector based on activity data (such as traffic counts). These profiles can be from different years
and countries, so that the full range of possibilities is included. These are compared to the fixed time profiles to
estimate the uncertainties, which are normally distributed (see Figure 3 for the uncertainty per aggregated GNFR
sector). We assume no error correlations exist.

**Table 3: Percentage (%) of emissions of $CO_2$ and CO (fossil + biofuel) that are attributed to large point sources (accounted for in databases) for source sectors public power and industry.**

| | $CO_2$ | | CO | |
|---|---|---|---|---|
| **Country** | **Public power** | **Industry** | **Public power** | **Industry** |
| **Netherlands** | 84.3 % | 80.4 % | 80.7 % | 86.0 % |
| **Belgium** | 65.4 % | 77.5 % | 99.5 % | 93.5 % |
| **Luxembourg** | 67.1 % | 67.2 % | 61.8 % | 94.2 % |
| **Germany** | 85.9 % | 74.1 % | 96.7 % | 87.9 % |
| **Czech Republic** | 89.2 % | 90.4 % | 79.3 % | 94.3 % |

**2.3 The Monte Carlo simulation**
Within a Monte Carlo simulation we create an ensemble (size N) of emissions, spatial proxies and time profiles
by drawing random samples from the covariance matrices described in Sect. 2.2. This creates a set of possible
solutions in the emission space, reflecting the uncertainties in the underlying parameters. The entire process is
shown in Figure 4. As mentioned before, not all sub-sectors are included in the Monte Carlo simulation and the
non-included emissions are added to each ensemble member at the final stage. Important is to ensure that the time





profiles and the spatial proxies do not affect the total emissions, so proxies should sum up to 1 for each country
and time profiles should be on average 1 over a full year. Before doing this, negative values are removed.
The source sectors that include point source emissions (mainly public power and industry) are treated separately.
The large point source emissions and their locations are relatively well-known and available from databases (e.g.
from E-PRTR), and therefore not included in the Monte Carlo. The remaining part of the emissions (non-point
source or small point sources) from these sectors are distributed using generic proxies (e.g. industrial areas) and
are calculated as the difference between the total emissions (activity data x emission factor) and the sum of the
point source emissions. If negative emissions result from this subtraction of reported point source emissions, the
residual is set to zero. Note that the spatial uncertainty of this residual part is often high. The fraction of the public
power and industrial emissions that are attributed to large point sources are shown in Table 3 for several countries.

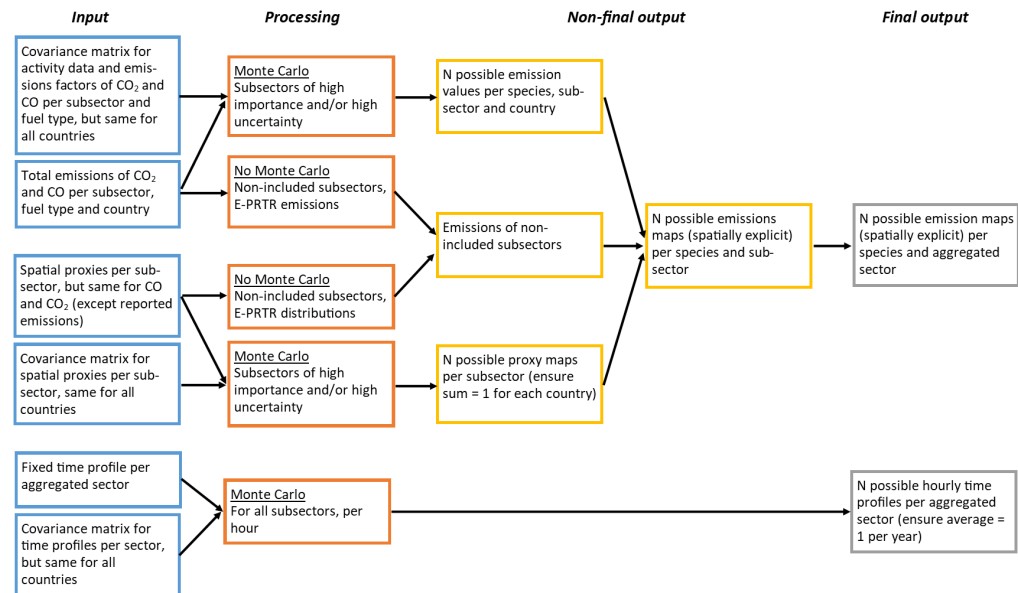


**Figure 4: Flow-diagram showing the input, processing and output of the Monte Carlo simulation.**
**2.4 Experiments to explore uncertainty propagation**
In this paper several experiments are performed to examine the impact of the uncertainties in different parameters
on the overall emissions and simulated concentrations:
1)   The first experiment uses a MC simulation (N=500) to illustrate the spread in emissions per sector due to

uncertainties in emission factors and activity data (no spatial/temporal variability is considered). This

experiment is used to show the contribution of specific sectors to the overall uncertainty and to illustrate

how uncertainties vary between sectors and countries. For this experiment country totals are used, also for

the countries that are partially outside the zoom domain shown in Figure 1. The results are presented in

Sect. 3.1.

2)   The second experiment uses a MC simulation (N=500) to illustrate how the uncertainty in spatial proxy

maps is translated into uncertainties in simulated concentrations (emissions are taken constant; no temporal

variability is included). We use emissions of other stationary combustion ($CO_2$) and road traffic (CO) to



illustrate the importance of having a correct spatial distribution for measurements close to the source area
and further away. The results are presented in Sect. 3.2.
3) The third experiment compares two spatial proxy maps for distributing 'residual' power plant emissions
(i.e. those not accounted for in point source databases) to illustrate the potential impact of spreading out
small point source emissions when zooming in on small case study areas (emissions are taken constant; no
temporal variability is included). The results are presented in Sect. 3.2.
4) The fourth experiment uses a MC simulation (N=500) to illustrate the spread in time profiles (emissions are
taken constant; no spatial variability is considered). We use this information to determine the error
introduced when extrapolating daytime (12–16 h LT) emissions (for example resulting from an inversion)
to annual emissions using an incorrect time profile. Figure 5 shows two possible daily cycles, which have
46 % (blue) and 25 % (orange) of their emissions between 12 and 16 h LT. Therefore, both time profiles
will give a different total daily emission when used to extrapolate the daytime emissions. The results are
presented in Sect. 3.3.

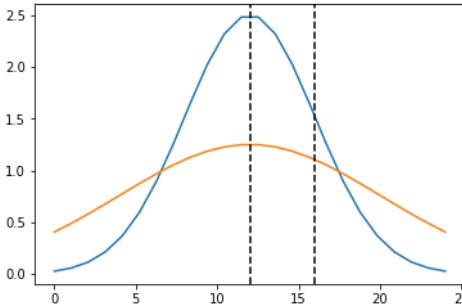


**Figure 5: Schematic overview of two possible time profiles, which represent a different fraction of the total daily emissions during the selected period (12–16 h LT, illustrated by the dashed lines).**

5) For the final experiment, maps are made of the (absolute and relative) uncertainty in each pixel, including
uncertainties in emission factors, activity data and spatial proxies (no temporal variability). For this we used
a Tier 1 approach, using the following equations:
$Total\ relative\ uncertainty = \sqrt{\sum standard\ deviations^2}/emission\ sum$    (1)
for the summation of uncorrelation quantities (e.g. sectoral emissions), and:
$Total\ relative\ uncertainty = \sqrt{\sum relative\ uncertainties^2}$    (2)
for the multiplication of random variables, such as used to combine activity data and emission factors. Here,
the (total) relative uncertainty is the percentage uncertainty (uncertainty divided by the total) and the
standard deviations are expressed in units of the uncertain quantity (percentage uncertainty multiplied with
the uncertain quantity). These maps are used to explore spatial patterns in uncertainties and examine what
we can learn about different countries or regions. The results are presented in Sect. 3.4.
For experiment 2 and 3 a smaller domain is selected to represent a local case study (Figure 6). We used the
Rotterdam area, which has been studied in detail before (Super et al., 2017b, 2017a). The domain is about 34x26
km and centred over the city, which includes some major industrial activity as well. To translate the emissions
into atmospheric concentrations, a simple plume dispersion model is used, the Operational Priority Substances



(OPS) model. This model was developed to calculate the transport of pollutants, including chemical
transformations (Van Jaarsveld, 2004; Sauter et al., 2016) and was adapted to include CO and $CO_2$ (Super et al.,
2017a). The short-term version of the model calculates hourly concentrations at specific receptor points,
considering hourly variations in wind direction and other transport parameters. Although the model is often used
for point source emissions, it can also handle surface area sources. This model was chosen because of its very
short run time, which makes it suitable for a large ensemble.

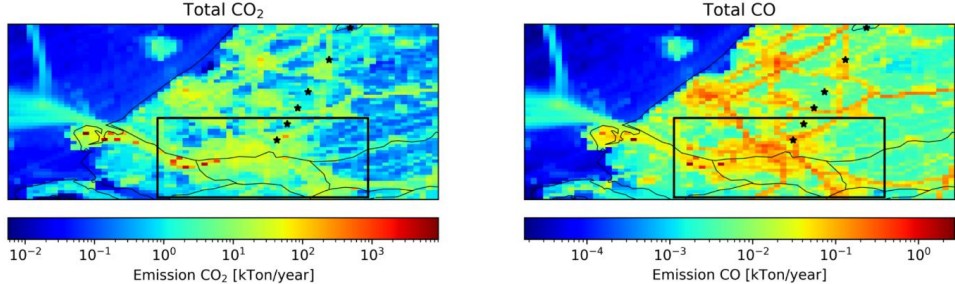


**Figure 6: Emissions of $CO_2$ and CO for part of the Netherlands, including the sub-domain (black rectangle) over**
**Rotterdam. Black stars indicate the receptor locations.**
The OPS model is run for each ensemble member for 5 January 2014 from the start of the day until 16 h LT. On
this day the wind direction is relatively constant at about 215° and the wind speed is around 6 m s$^{-1}$. We specify
receptor points downwind from the centre of our domain at increasing distance (5, 10, 15, 20, 30 and 40 km). We
use the last hour of the simulation for our analyses. We assume emissions from other stationary combustion and
road traffic (experiment 2) to take place at the surface. The initial emissions of 'residual' power plants, smeared
out over all industrial areas, are also emitted at the surface. However, we raise the height of the emissions to 20m
when these emissions are appointed to specific pixels. This height is representative for stack heights of small
power plants.
**3 Results**
**3.1 Uncertainties in total emissions**
Using the uncertainties in emission factors and activity data we can evaluate the uncertainty in the total emissions
of $CO_2$ and CO per sector. Figure 7 shows the normalized spread in emissions per sector. The $CO_2$ emissions have
a relatively small uncertainty range and the uncertainty in the total emissions (all sectors together) is only about 1
% (standard deviation). The largest uncertainties are for fugitives and aviation, which are only small contributors
to the total $CO_2$ emissions (1.3 % and 0.4 %, respectively). Therefore, their contribution to the total emission
uncertainty is very small, as is shown in Figure 8. The largest uncertainty in the total $CO_2$ emissions is caused by
the public power sector. Despite the relatively small uncertainty in the emissions from this sector, it is the largest
contributor to the total $CO_2$ emissions (33 %) and therefore the uncertainty in the public power sector contributes
about 45 % to the uncertainty in the total $CO_2$ emissions.
In contrast, the CO emissions show a larger uncertainty bandwidth with many high outliers caused by the
lognormal distribution of uncertainties in the emission factors. The uncertainty in the total emissions (all sectors
together) is 6 % for CO (standard deviation). Here, again the largest uncertainties are related to sectors (public




305 power and road transport (LPG fuel)) that are relatively small contributors to the total CO emissions. The main

306 contributor to the uncertainty in total CO emissions is other stationary combustion, which contributes about 31 %

307 to the total emissions and is responsible for more than 60 % of the total uncertainty.

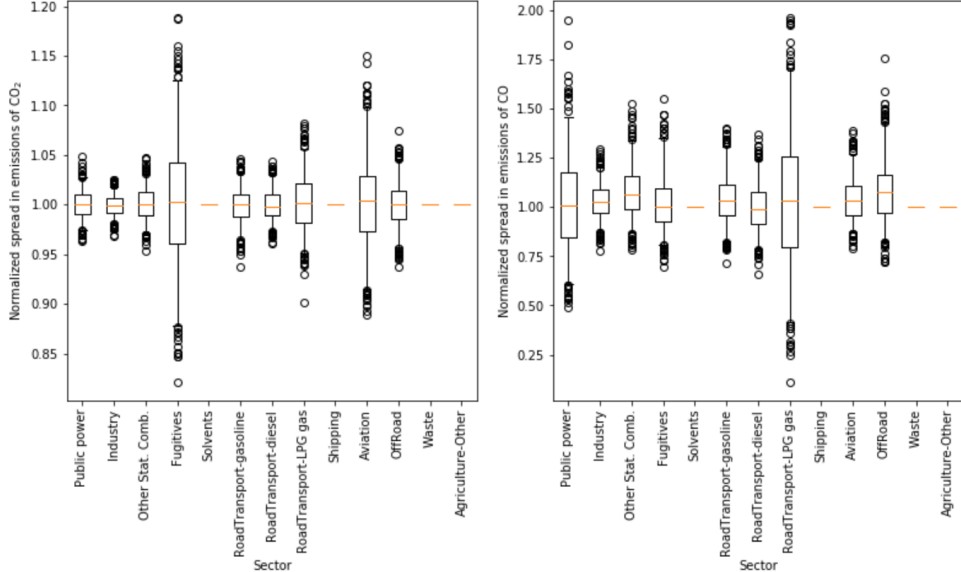


**Figure 7: Normalized spread in emissions of $CO_2$ and CO. The box represents the interquartile range, the whiskers the**
**2.5–97.5 percentile, the lines the median values, and the circles are outliers. For sectors where no box is drawn there is**
**no data included in the Monte Carlo simulation. Note the different scales of the y-axis.**

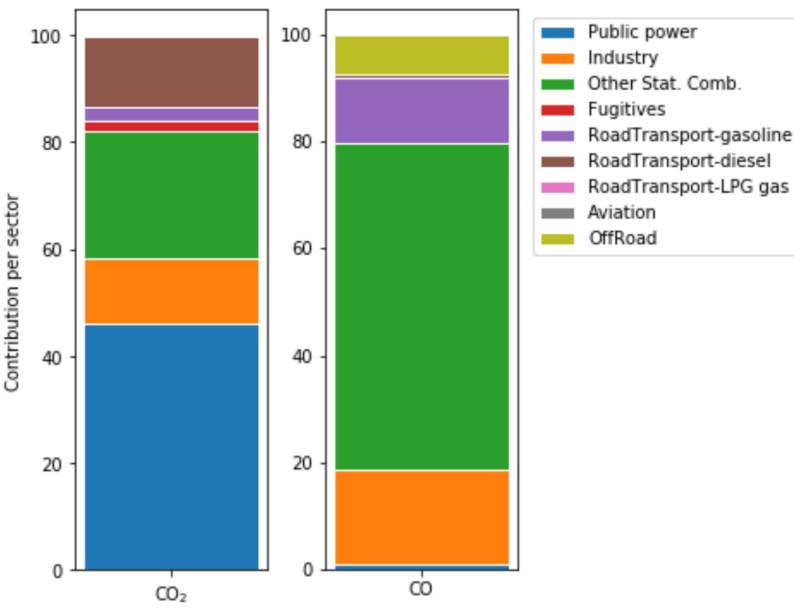


**Figure 8: Contribution of source sectors to the total uncertainty in $CO_2$ and CO emissions, summing to 100 %.**





Although the uncertainty in each parameter is assumed to be the same for each country, how a sector is composed
of sub-sectors can vary per country. Therefore, the uncertainty per aggregated sector can also vary per country.
An example is shown in Figure 9 (left panel), which shows the normalized spread in $CO_2$ emissions of other
stationary combustion for all countries within the domain. We find a much larger uncertainty in countries where
the relative fraction of biomass combustion is larger, because biomass burning has a much larger uncertainty in
both the activity data and the emission factor. For example, the percentage of biomass burning in the residential
sector is 54 % for the Czech Republic and 65 % for Denmark, compared to only 11 % and 9 % for the Netherlands
and Great Britain. Thus, differences in the fuel composition of countries result in differences in the overall
emission uncertainties, even if the uncertainty per parameter is estimated to be the same. Overall, the differences
between countries are relatively small (Figure 9, right panel).

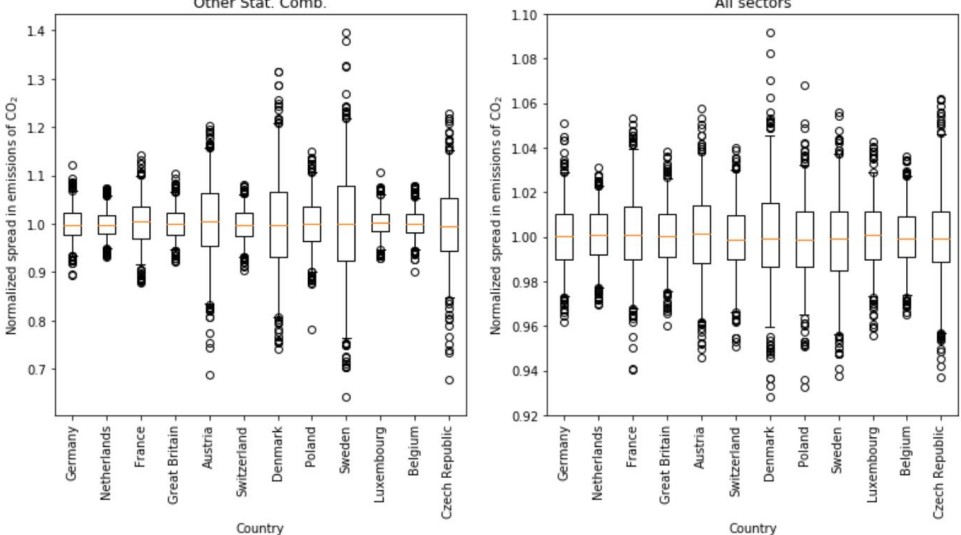


**Figure 9: Normalized spread in emissions of $CO_2$ for other stationary combustion and all sectors combined for a range**
**of countries. The box represents the interquartile range, the whiskers the 2.5–97.5 percentile, the lines the median**
**values, and the circles are outliers.**
**3.2 Uncertainties in spatial proxies**
We examined the impact of uncertainties in spatial proxies on modelled $CO_2$ and CO concentrations for major
source sectors. For $CO_2$ we selected other stationary combustion (only commercial/residential, no
agriculture/forestry/fishing). The largest fraction (>90 %) of $CO_2$ emissions from this sector is distributed using
population density as proxy, which is used here (the remainder of the emissions is not considered). The uncertainty
in this sector-proxy combination is estimated to be 50% (normal distribution), mainly due to the disaggregation
to the 1x1 km resolution. For CO we selected road transport (all fuels, but only passenger cars). The spatial proxy
for distributing passenger car emissions is based on traffic intensities compiled using Open Transport Map and
Open Street Map, vehicle emission factors per road type/vehicle type/country, and fleet composition. The
uncertainty in this proxy is estimated to be 30 % (normal distribution) due to a higher intrinsic resolution.
Figure 10 shows the resulting spread in atmospheric concentrations as a function of downwind distance from the
source area. For $CO_2$ (left panel) we see a concentration of about 3.0 ppm at 10 km from the source area centre,



but with a large uncertainty bandwidth. This signal is large enough to measure, but with this large uncertainty
such measurements are difficult to use in an inversion. The measurement at 5 km from the source area centre is
slightly lower than the one at 10 km, because it is downwind of a part of the emissions. At longer distances, the
magnitude of the atmospheric signal decreases drastically, and so does the absolute spread in concentrations. The
signal becomes too small compared to the uncertainties occurring in a regular inversion framework to be useful.
The right panel shows a similar picture for the CO concentrations resulting from passenger car emissions. Again,
the spread in concentrations is large close to the source area centre and decreases with distance, but also the
magnitude of the signal decreases. Note that we focus here on a single source sector and the overall signals will
be larger and therefore easier to use. Nevertheless, the large spread in concentrations shows that a good
representation of the spatial distribution is important for constraining sectoral emissions.

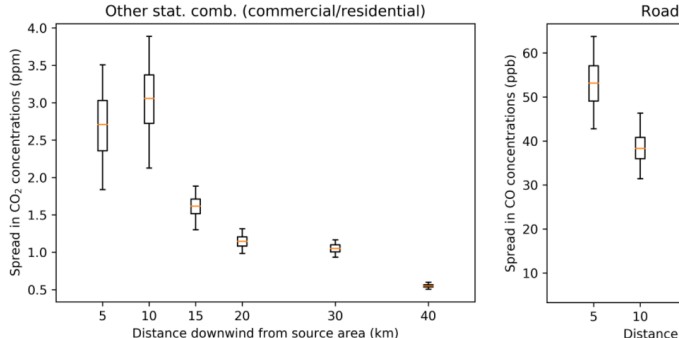
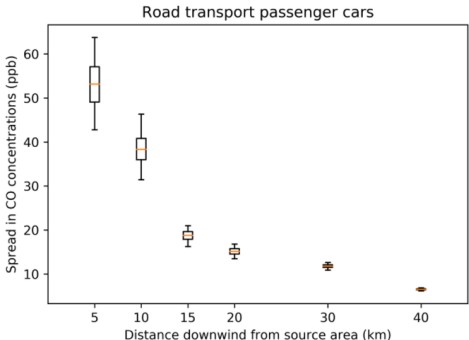


**Figure 10: Spread in concentrations of $CO_2$ resulting from commercial/residential emissions due to uncertainties in the**
**total population proxy map and spread in concentrations of CO resulting from road transport (passenger cars)**
**emissions due to uncertainties in the passenger cars proxy map. The box represents the interquartile range, the**
**whiskers the 2.5–97.5 percentile, and the lines the median values.**
Both proxy maps discussed here are the main proxy maps for the selected sectors. As mentioned before, some
sectors have residual emissions that are distributed using an alternative proxy map. An example is public power.
Large power plants are listed in databases, including the reported emissions (Table 3). The remainder of the
country emissions is spatially distributed over all industrial areas. However, it is more likely that the residual
emissions should be attributed to specific point sources (small power plants not listed in databases). That means
that instead of spreading the emissions over a large area, leading to very small local emissions and a low
concentration gradient, there could be relatively large emissions at a few locations. Therefore, the uncertainty in
these sector-proxy combinations is assumed to have a lognormal distribution, in part because of the absence of a
better estimation.
We illustrate the effect of this assumption by creating a new proxy map for residual (small) power plants. We find
that for the Netherlands a total capacity of 3655 MWe by 676 combustion plants is not included as a point source
(source: S&P Global Platts World Electric Power Plants database (https://www.spglobal.com/platts/en/products-
services/electric-power/world-electric-power-plants-database)). At least 70 % of this capacity, attributed to 280
plants, is assumed to be in industrial areas. Given 4052 grid cells designated as industrial area in the Netherlands,
this is just 7 % of the total amount of industrial area grid cells assuming no more than one plant per grid cell. The
remainder is mainly related to cogeneration plants from glasshouses, which are located outside the industrial areas.
Therefore, we create a new proxy map for power plants by equally assigning 70 % of the emissions from the
residual power plants to 20 randomly chosen pixels (7 % of the total amount of industrial area pixels in the case
study area, i.e. the same density as for the Netherlands as a whole). As mentioned before, we also raise the height
of the emissions from surface level to 20 m, which is a better estimate of the stack height for small power plants.
The effect on local measurements is large (Figure 11). Instead of measuring a small and constant signal from this
sector, the assumed presence of small power plants results in measuring occasional large peak concentrations.
Thus, despite being relatively unimportant at the national level, for local studies the impact of the uncertainty in
these 'residual' proxies can be large.

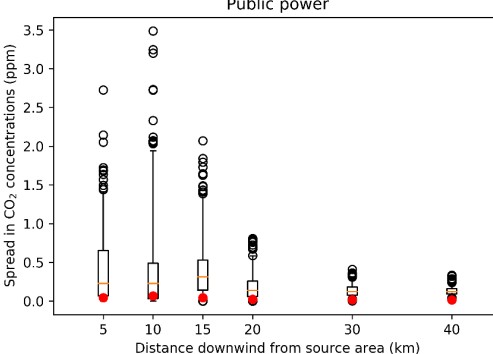


**Figure 11: Spread in concentrations of $CO_2$ resulting from public power emissions due to differences in the proxy map:**
**emissions are distributed using the new proxy map with only 20 randomly chosen pixels containing emissions. The box**
**represents the interquartile range, the whiskers the 2.5–97.5 percentile, the lines the median values, and the black**
**circles are outliers. The red dots show concentrations of $CO_2$ when the original proxy map is used (industrial area).**
**3.3 Uncertainties in time profiles**
The timing of emissions is important to interpret measurements correctly. During morning rush hour, a peak is
expected in road traffic emissions, but the magnitude of this peak can differ from one day to the next. Also, the
seasonal cycle in emissions due to heating of buildings can vary between years due to varying weather conditions.
Yet, often fixed time profiles are used to describe the temporal disaggregation of annual emissions. The range of
possible values for the time profile of other stationary combustion is shown in Figure 12. The range can be very
large, especially during the winter. However, note that the average of each time profile is 1.0 for a full year, so
that the temporally distributed emissions add up to the annual total. Therefore, changes in the time profile indicate
shifts in the timing in the emissions and not changes in the overall emissions due to cold weather, which are
accounted for by the activity data.

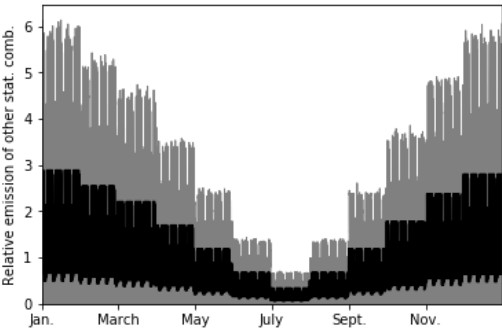


**Figure 12: Spread in time profiles for other stationary combustion (N=500), resulting from the Monte Carlo simulation.**
**The black line represents the standard time profile.**

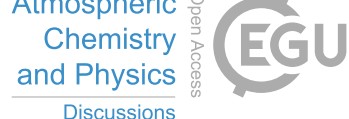



In inverse modelling, often well-mixed (non-stable) daytime measurements are selected, because these are least
prone to errors in model transport. The total annual emissions can then be calculated using a time profile. However,
if the time profile is not correct, an incorrect fraction of the emissions can be attributed to the selected hours. We
examined the impact of using an incorrect time profile on the total yearly emissions by calculating yearly
emissions for each ensemble member. Figure 13 shows the normalized spread in sectoral emissions for all
ensemble members. The error in the total annual emissions, resulting from the upscaling of daytime emissions
using an incorrect time profile, can reach up to about 1–2 %. This is a significant source of error for country-level
$CO_2$ emissions, but less important for CO as the other uncertainties for CO are much larger.

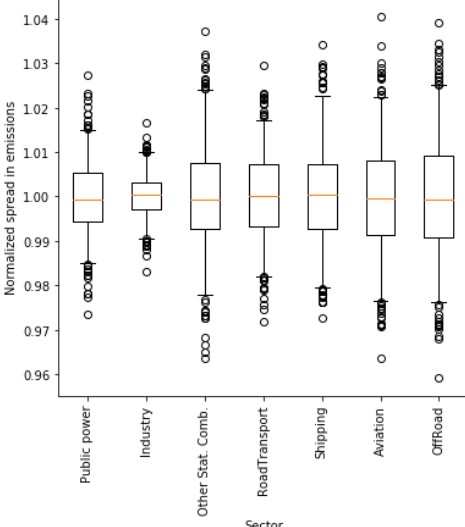


**Figure 13: Normalized spread in emissions per sector due to uncertainties in time profiles. The box represents the**
**interquartile range, the whiskers the 2.5–97.5 percentile, the lines the median values, and the circles are outliers.**
**3.4 Uncertainty maps and spatial patterns**
As mentioned before, the uncertainty of the emission value in a grid cell is determined by the uncertainties in
activity data, emission factors and spatial distribution proxies. The gridded uncertainty maps in Figure 14 and
Figure 15 illustrate that countries or (types of) regions differ significantly in their emission uncertainty, both in
absolute and relative values. Concerning the uncertainty in $CO_2$ and CO emissions, several observations can be
made.
First, for both CO and $CO_2$ the road network is visible due to low relative uncertainties and high absolute
uncertainties compared to the surroundings. This indicates that, despite having large emissions per pixel, the
spread in road traffic emissions among ensemble members is relatively small. This is likely due to the small
(normally distributed) uncertainty in the spatial proxies for road traffic, i.e. the location of the roads is well-known.
The surrounding rural areas are dominated by other stationary combustion, which has a slightly larger spatial
uncertainty.

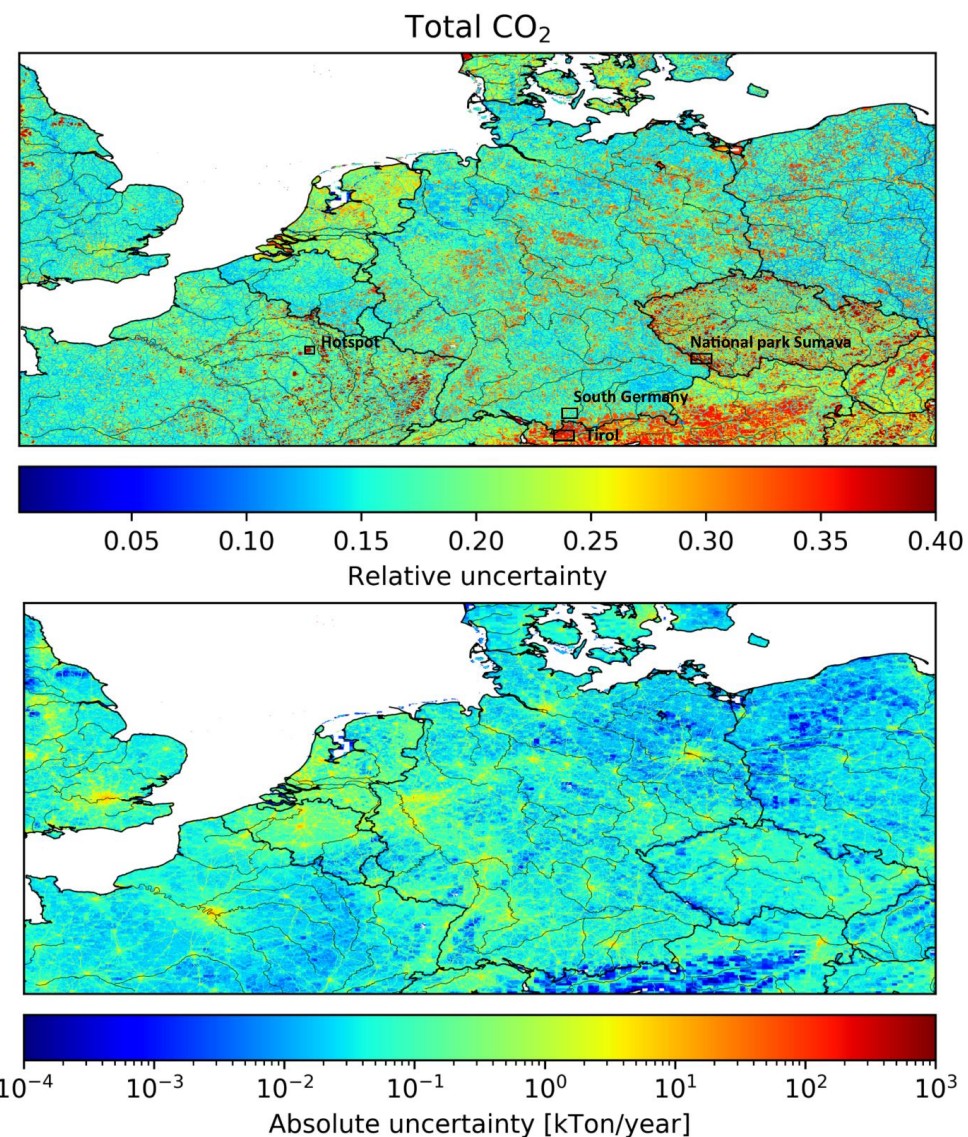


**Figure 14: Maps of the relative and absolute uncertainty in CO₂ emissions. Areas that are examined in more detail are outlined by black squares in the top panel.**

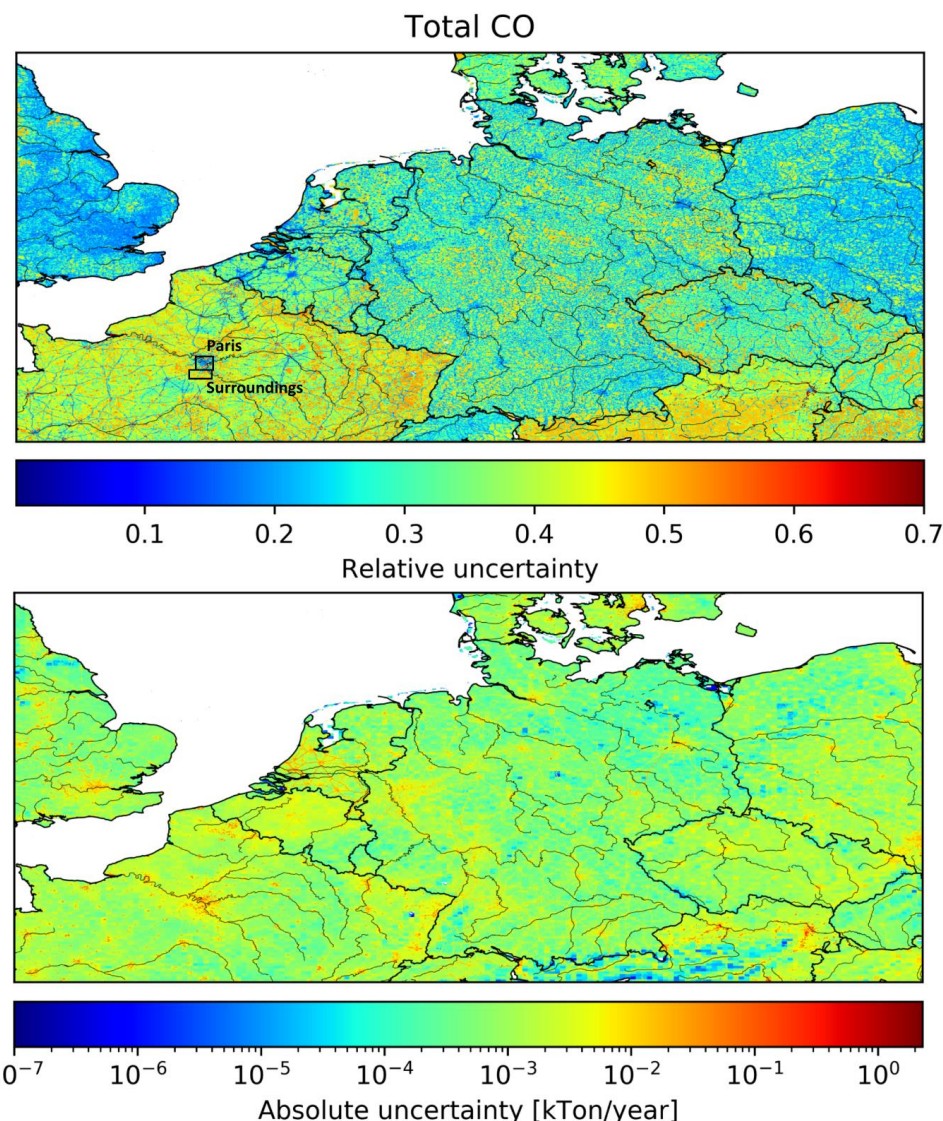

**Figure 15: Maps of the relative and absolute uncertainty in CO emissions. Areas that are examined in more detail are outlined by black squares in the top panel.**

Second, in Austria (Tirol mainly) a large area of high relative uncertainty in $CO_2$ emissions is visible (average pixel emission is 220 tonnes $CO_2$ $yr^{-1}$), which we compare to an area just on the other site of the border in southern Germany (average pixel emission is 495 tonnes $CO_2$ $yr^{-1}$). The uncertainty in both areas is dominated by other stationary combustion. Yet, in Austria a lot of biofuels are used (52 % of the total emissions for this source sector) with a large uncertainty in the emission factor and spatial distribution, whereas in Germany only 20 % of the emissions in this sector are caused by biofuels. On the other hand, the absolute uncertainty is very small in Tirol because of the low population density (and thus small emissions) in this mountainous area.



Third, some large patches of high relative uncertainty in $CO_2$ emissions are visible in the Czech Republic and northeast France. The location of these patches seems to correspond to natural areas/parks. Therefore, absolute uncertainties are low in these areas given the low emissions (average pixel emission in the Sumava national park is 22 tonnes $CO_2$ yr$^{-1}$). The total uncertainty can be explained for 60 % by the uncertainty in other stationary combustion, mainly wood burning (Figure 16). Also, agriculture (field burning of residues) plays a significant role. In addition to these natural areas, there are also some very small dark red areas (relative uncertainty) in northern France. These areas are military domain and have a lower absolute uncertainty than their surroundings because very few emissions are distributed to these areas (average pixel emission is 250 tonnes $CO_2$ yr$^{-1}$). The public power and industrial emissions are probably too small to be reported, hence the large relatively uncertainty.

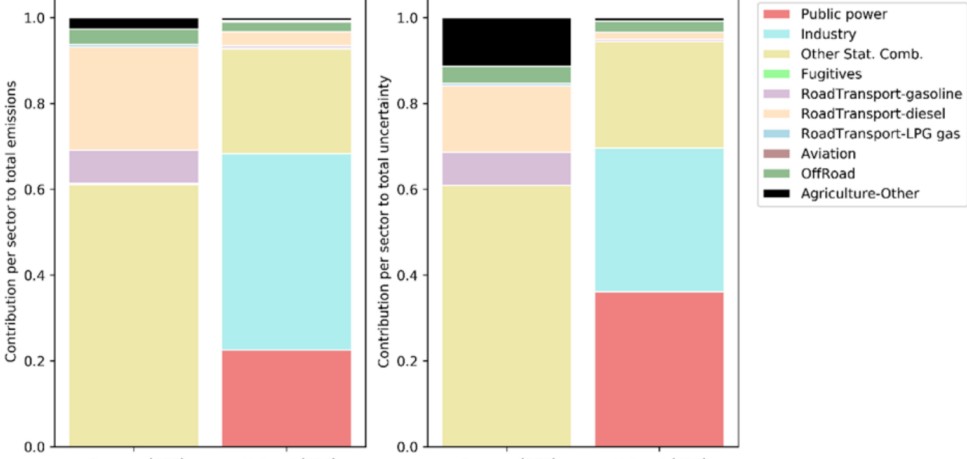

**Figure 16: Contribution of source sectors to the total emissions (left) and the total uncertainty (right) in $CO_2$ for the Sumava national park in the Czech Republic and a hotspot in France, summing to 100 %. See Figure 14 for the exact location of these areas.**

Fourth, big cities like Paris, Berlin and Brussels are clearly visible as areas where the relative uncertainty is lower than in the surrounding areas. Compared to its surroundings, the uncertainty in Paris is mainly determined by the industrial sector (Figure 17). Since industrial emissions are relatively well-known, the relative uncertainty is small. However, the absolute uncertainty is large for big cities because of the high emissions in these densely populated areas (average pixel emission is 64 tonnes CO yr$^{-1}$ for Paris). In the surrounding areas the emissions are again dominated by other stationary combustion, which has a larger uncertainty. Yet, the absolute uncertainty is smaller because of the lower emissions (average pixel emission is 12 tonnes CO yr$^{-1}$).



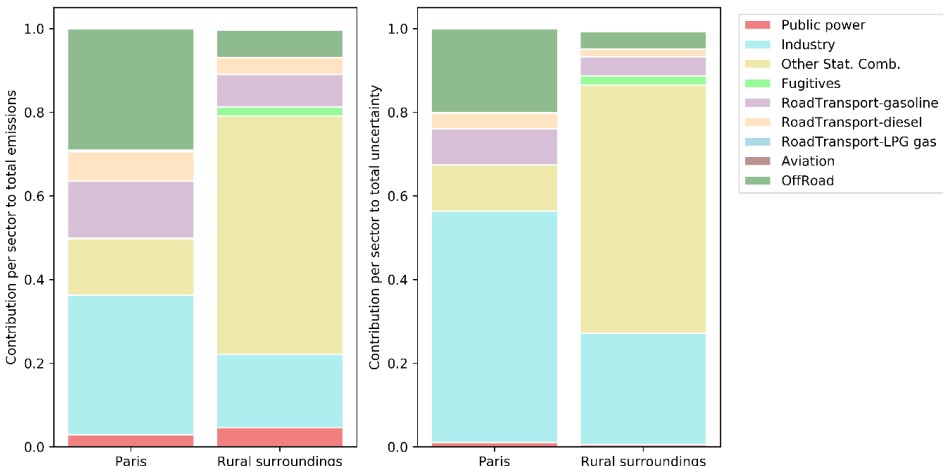

454

**Figure 17: Contribution of source sectors to the total emissions (left) and the total uncertainty (right) in CO for Paris and its surroundings, summing to 100 %. See Figure 15 for the exact location of these areas.**

Finally, the relative uncertainties seem to be consistently higher in some countries than in others. For example, the relative uncertainty in the total emissions of France and Great Britain (only pixels within the domain) are 39 % and 25 %, respectively. For France, the main sources of uncertainty are industry and other stationary combustion, whereas the off-road and road transport sectors have a significant contribution to the uncertainty in Great Britain (Figure 18). The main difference between the countries is again the amount of biomass used in the other stationary combustion sector (26 % in France and 8 % in Great Britain). This is likely to explain why in rural areas the relative uncertainty is much higher in France.

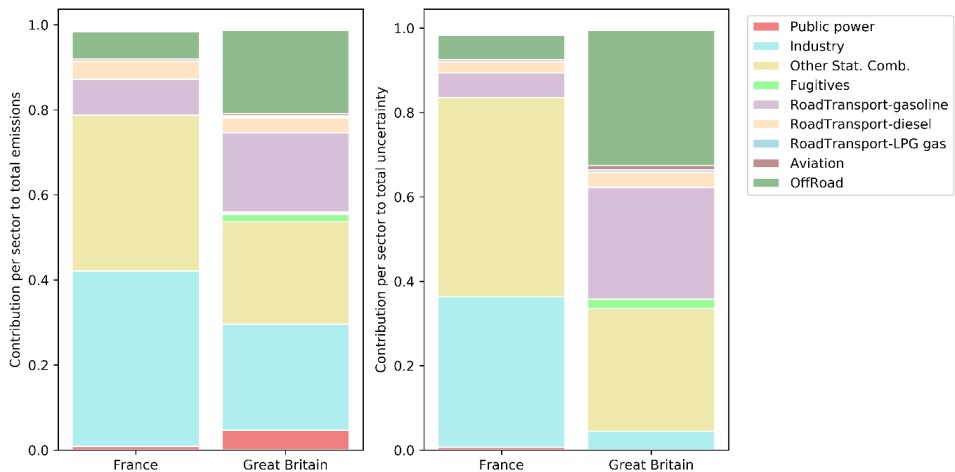

464

**Figure 18: Contribution of source sectors to the total emissions (left) and the total uncertainty (right) in CO for France and Great Britain, summing to 100 %.**



**4. Discussion and conclusions**

Several previous studies have examined the uncertainty in emissions, either globally or nationally. For example, Andres et al. (2014) studied the uncertainty in the CDIAC emission inventory on a global scale, suggesting that the largest uncertainties are related to the fuel consumption (i.e. activity data). A similar concern was identified for China, for which the uncertainty in energy statistics result in an uncertainty ratio of 15.6 % in the 2012 $CO_2$ emissions (Hong et al., 2017). In the present study the uncertainties in activity data and emission factors are similar for $CO_2$, whereas the uncertainty in CO emission factors is much larger than the uncertainty in activity data. In addition, many countries report uncertainties in emission estimates in their National Inventory Reports (UNFCCC, 2019). Yet, their methods differ and can even vary over time. Several scholars have examined the uncertainty in national $CO_2$ emissions in more detail. For example, Monni et al. (2004) (Finland) and Fauster et al. (2011) (Denmark) used a Tier 2 approach (Monte Carlo simulation) to determine the uncertainty in the total greenhouse gas emissions. They found an uncertainty of about 5–6 % for the year 2001 for Finland and an uncertainty of 4–5 % for the year 2008 for Denmark, also considering non-normal distributions in uncertainties. Moreover, Oda et al. (2019) found a 2.2 % difference in total $CO_2$ emissions in Poland between two emission inventories. These values agree with our total emission uncertainties.

Even fewer studies have focused on uncertainties in the proxy maps used for spatial disaggregation. Some studies compared emission inventories to get an idea of the spatial uncertainties (Gately and Hutyra, 2017; Hutchins et al., 2017), but these studies are likely to underestimate uncertainties due to systematic errors caused when different emission inventories use similar methods and/or proxies for spatial allocation. Moreover, exact quantification of uncertainties is often limited, dependent on the spatial scale, and the uncertainties are not specified per source (i.e. total emissions and spatial disaggregation) (Oda et al., 2019). Sowden et al. (2008) used a qualitative approach to identify the uncertainty of different components of their emission inventory for reactive pollutants (activity, emission factors, spatial and temporal allocation and speciation) by giving each component a quality rating. They suggest that spatial allocation is an important source of uncertainty for residential burning, but not so much for point sources and road traffic. Indeed, the location of large point sources and roads is relatively well-known. However, we consider the allocation of emissions to pixels that include roads to have a significant (pixel value) uncertainty. Therefore, our results show that uncertainties in the spatial proxy used for road traffic can cause a significant spread in CO concentrations.

Andres et al. (2016) did a more extensive analysis of the spatial distribution in CDIAC, including uncertainties in pixel values (e.g. due to incorrect accounting methods or changes over time) and due to the representativeness of the proxy for the spatial distribution of emissions (also see Sect. 2.2.3). We considered these sources of uncertainty as well. However, Andres et al. (2016) also mention spatial discretization as a source of error, because they assign each pixel (1x1° resolution) to one country. The proxy maps used in this study include country fractions in each pixel, reducing this uncertainty. In contrast, we suggest another source of uncertainty, namely the fact that some pixels can include emissions while no activity takes place there or vice versa (proxy quality). Based on the listed uncertainties, Andres et al. (2016) found an average uncertainty ($2\sigma$) in individual pixels of 120 % (assuming normal distributions). Here, we find an average uncertainty ($2\sigma$) of 36 %. However, a small number of large outliers occurs (less than 0.01 % of the pixels has an uncertainty of >1000 %) due to lognormal error distributions, although these are related to pixels with small emissions. A large part of the difference can be explained by the large pixel size of CDIAC and the large error introduced by spatial discretization (e.g. due to pixels that cover





507   large areas of two different countries). Also, their emissions are spatially distributed based on population density,

508   while we use a range of proxy maps depending on the source sector and use specific locations for large point

509   sources. However, the uncertainty estimates are partially based on expert judgement and remain subjective.

510   Moreover, the uncertainty related to the location of actual activities is not included in our uncertainty estimate,

511   even though we have shown this can have a large impact locally.

512   The country-level $CO_2$ emissions used for our emission inventory are based on NIR's, which are assumed to be

513   relatively accurate because of the use of detailed fuel consumption statistics and country-specific emission factors

514   (Andres et al., 2014; Francey et al., 2013). The uncertainties reported in the NIRs were determined following

515   specified procedures and are deemed the most complete and reliable estimates available. Yet, because of the use

516   of prescribed methods and in some cases general emission factors, systematic errors can occur both in the estimate

517   of parameters and in the estimate of uncertainties. We choose to average the uncertainties reported by several

518   countries, because the uncertainty estimates are relatively consistent across countries. However, this would not

519   eliminate such systematic errors. The effect of systematic errors could be analysed by comparing different sources

520   of information. Additionally, we assume point source emissions are relatively certain, yet a recent study showed

521   that significant uncertainties exist in reported emissions of US power plants (Quick and Marland, 2019). A similar

522   study for Europe is recommended, not only to improve the knowledge for the European situation, but also to

523   understand continental differences.

524   One source of uncertainty that is not considered in this study is the incompleteness of the emission inventory (i.e.

525   if sources are missing) or double-counting errors. For example, during the compilation of the base inventory we

526   found that in several cases the $CO_2$ emissions from airports were very low. The reason was that emissions from

527   international flights are not reported in the NIR's and therefore not part of the emission data used to create the

528   inventory. Once discovered, this was corrected and LTO (aircraft landing and take-off) emissions from

529   international flights were added in a later stage. Such discrepancies caused by reporting guidelines could be

530   present for other source types as well. Although overall this error is likely to be small, locally the errors might be

531   significant.

532   Finally, Sowden et al. (2008) mention (dis)aggregation as another source of error, i.e. the calculation of emissions

533   on a different scale (spatially, temporally or sector level) than the input data. In principle, fuel consumption data

534   is available on aggregated levels and then separated over different subsectors. This increases the uncertainty at

535   the lower level, but on the aggregated level the uncertainties remain the same. A similar note was made by Andres

536   et al. (2016) about the use of higher resolution proxy maps, which might increase the uncertainty due to lack of

537   local data. However, when local data is available this might also decrease the uncertainties. For example, the

538   EDGAR emission database uses non-country specific emission factors based on technology levels and sector

539   aggregated energy statistics (Muntean et al., 2018). The reason is that the level of detail we used in this paper is

540   not available globally. However, using generic emission factors can introduce large uncertainties when sub-

541   sectoral chances occur. Therefore, regional/local studies could benefit from using a dedicated emission inventory

542   for their region of interest instead of a global inventory.

543   In this work we studied the uncertainties in a high-resolution gridded emission inventory for $CO_2$ and CO,

544   considering uncertainties in the underlying parameters (activity data, emission factors, spatial proxy maps and

545   time profiles). We find that all factors play a significant role in determining the emission uncertainties, but that

546   the contribution of each factor differs per sector. Disaggregation of emissions introduces additional sources of



uncertainty, which makes uncertainties at higher resolution larger than at the scale of a country/year and can have
a large impact on (the interpretation of) local measurements. This is an important consideration for inverse
modelers and our methodology can be used to better define local uncertainties for e.g. urban inversions. Inverse
modelers should be aware that the use of erroneous time profiles to extrapolate emission data could result in errors
of a few percent. Moreover, we found that large regional differences exist in absolute and relative uncertainties.
By looking in more detail at specific regions (or countries) more insight can be gained about the emission
landscape and what are the main causes of uncertainty. Interestingly, areas with larger absolute uncertainties often
have smaller relative uncertainties. A likely explanation is that large sources of $CO_2$ and CO emissions received
more attention and are therefore relatively well-constrained, for example in the case of large point sources.
Nevertheless, since we are most interested in absolute emission reductions the map with absolute uncertainties
can be used to define an observational network that is able to reduce the largest absolute uncertainties. Finally,
we believe that an uncertainty product based on a well-defined, well-documented and systematic methodology
could be beneficial for the entire modelling community and support decision-making as well. However, specific
needs can differ significantly between studies, for example the scale/resolution, source sector aggregation level,
and which species are included. Therefore, the creation of a generic uncertainty product is challenging and needs
further research.
**Data availability**
The emission inventories are available for non-commercial applications and research. Please contact Hugo Denier
van der Gon (hugo.deniervandergon@tno.nl).



## Appendix A

**Table A1: Relative uncertainties (fraction) in activity data and CO₂ emission factors as taken from the NIRs (country-average) and in CO emission factors as derived from literature (assumed equal for all countries in the domain). The quoted uncertainty ranges are assumed to be representative for one standard deviation. Uncertainties in activity data and CO₂ emission factors are often relatively low and symmetrically distributed and normal distributions (Norm) are assumed for these activities. Compared to CO₂ emission factors, the uncertainty in CO emission factors is much higher, up to an order of magnitude. Uncertainties in CO emission factors are often lognormally distributed (Logn) and are assumed equal for all countries in the HR domain. The uncertainty in the activity of open burning of waste (not covered by the NIRs) is also assumed to have a lognormal distribution.**

| Sector (NFR) | Fuel type | Activity data | | CO₂ emission factors | | CO emission factors | |
|---|---|---|---|---|---|---|---|
| | | Average | Distribution | Average | Distribution | Average | Distribution |
| Public electricity and heat production (1.A.1.a) | Solid (fossil) | 0.018 | Norm | 0.030 | Norm | 0.149 | Logn |
| | Liquid (fossil) | 0.022 | Norm | 0.031 | Norm | 0.399 | Norm |
| | Gaseous (fossil) | 0.021 | Norm | 0.015 | Norm | 0.513 | Norm |
| | Biomass | 0.060 | Norm | 0.05 | Norm | 0.231 | Logn |
| Oil and gas refining (1.A.1.b & 1.B.2.d) | All | 0.038 | Norm | 0.048 | Norm | 0.402 | Norm |
| Oil production & Gas exploration (1.B.2 mainly flaring, 1.B.2.c) | All | 0.118 | Norm | 0.141 | Norm | 0.240 | Logn |
| Iron and steel industry (1.A.2.a & 2.C.1) | All | 0.044 | Norm | 0.056 | Norm | 0.240 | Logn |
| Non-ferrous metals (1.A.2.b & 2.C.2_3) | All | 0.031 | Norm | 0.029 | Norm | 0.208 | Norm |
| Chemical industry (1.A.2.c & 2.B) | All | 0.042 | Norm | 0.041 | Norm | 0.138 | Logn |
| Pulp and paper industry (1.A.2.d) | All | 0.027 | Norm | 0.016 | Norm | 0.138 | Logn |
| Food processing, beverages and tobacco (1.A.2.e) | All | 0.029 | Norm | 0.017 | Norm | 0.138 | Logn |
| Non-metallic minerals (1.A.2.f & 2.A) | All | 0.032 | Norm | 0.041 | Norm | 0.384 | Logn |
| Other manufacturing industry (1.A.2.g) | All | 0.029 | Norm | 0.014 | Norm | 0.138 | Logn |
| Civil aviation - LTO (1.A.3.a) | All | 0.089 | Norm | 0.040 | Norm | 0.231 | Logn |
| Road transport (all vehicle types) (1.A.3.b) | Gasoline (fossil) | 0.031 | Norm | 0.025 | Norm | 0.284 | Logn |
| | Diesel (fossil) | 0.032 | Norm | 0.026 | Norm | 0.319 | Norm |
| | Gaseous (fossil) | 0.039 | Norm | 0.027 | Norm | 0.320 | Logn |
| | LPG | 0.039 | Norm | 0.027 | Norm | 0.462 | Norm |
| Other transport (1.A.3.e & 1.A.4 mobile) | All | 0.067 | Norm | 0.023 | Norm | 0.384 | Logn |
| Other mobile (1.A.5.b) | All | 0.098 | Norm | 0.026 | Norm | 0.384 | Logn |
| Residential (1.A.4.b) | Gaseous (fossil) | 0.040 | Norm | 0.022 | Norm | 0.141 | Logn |
| | Liquid (fossil) | 0.048 | Norm | 0.024 | Norm | 0.404 | Norm |
| | Solid (fossil) | 0.085 | Norm | 0.041 | Norm | 0.141 | Logn |
| | Biomass | 0.163 | Norm | 0.055 | Norm | 0.384 | Logn |
| Commercial institutional (1.A.4.a) | Gaseous (fossil) | 0.043 | Norm | 0.022 | Norm | 0.138 | Logn |
| | Liquid (fossil) | 0.055 | Norm | 0.023 | Norm | 1.065 | Norm |
| | Solid (fossil) | 0.087 | Norm | 0.040 | Norm | 0.994 | Norm |
| | Biomass | 0.103 | Norm | 0.055 | Norm | 0.730 | Logn |


| | | | | | | | |
|---|---|---|---|---|---|---|---|
| Agriculture/Forestry/Fishing (1.A.4.c) | Gaseous (fossil) | 0.050 | Norm | 0.028 | Norm | 0.138 | Logn |
| | Liquid (fossil) | 0.051 | Norm | 0.029 | Norm | 1.065 | Norm |
| | Solid (fossil) | 0.095 | Norm | 0.048 | Norm | 0.994 | Norm |
| | Biomass | 0.096 | Norm | 0.09 | Norm | 0.730 | Logn |
| Other stationary (1.A.5.a) | Gaseous (fossil) | 0.097 | Norm | 0.023 | Norm | 0.138 | Logn |
| | Liquid (fossil) | 0.084 | Norm | 0.021 | Norm | 1.065 | Norm |
| | Solid (fossil) | 0.103 | Norm | 0.033 | Norm | 0.994 | Norm |
| | Biomass | 0.180 | Norm | 0.04 | Norm | 0.730 | Logn |
| Agricultural waste burning (3.F) | - | 1.609 | Logn | 0.2 | Norm | 0.429 | Norm |
| Uncontrolled waste burning (5.C.2) | - | 1.609 | Logn | 0.5 | Norm | 0.366 | Logn |

**Table A2: Relative uncertainties (fractions) at cell level resulting from the spatial distribution. The values listed represent the (one standard deviation) uncertainty of the emission per cell due to uncertainty sources 2 and 3 as listed in Sect. 2.2.3. All values in the table below are based on expert quantification and inevitably include a considerable amount of subjectivity. The data should therefore be considered as a first order indication only. Note that the natural logarithm (Ln) of the uncertainty fraction is given in case uncertainty has a lognormal distribution.**

| Sector name | Proxy name | Distribution | Uncertainty |
|---|---|---|---|
| Public electricity and heat production; Chemical industry; Food processing, beverages and tobacco (comb); Food and beverages industry; Other non-metallic mineral production; Small combustion - Commercial/institutional – Mobile | CORINE_2012_Industrial_area | Logn | 2.2 |
| Solid fuel transformation; Iron and steel industry (comb); Iron and steel production; Pulp and paper industry (comb); Pulp and paper industry; Non-metallic minerals (comb); Cement production | CORINE_2012_Industrial_area | Logn | 3.7 |
| Other manufacturing industry (comb); Other industrial processes; Manufacturing industry - Off-road vehicles and other machinery | CORINE_2012_Industrial_area | Logn | 1.4 |
| Oil and gas refining (comb); Oil and gas refining | CORINE_2012_Industrial_area | Logn | 3.7 |
| | TNO_PS for Refineries | Logn | 1.7 |
| Coal mining (comb) | CORINE_2012_Industrial_area | Logn | 4.6 |
| | TNO_PS for Coal mining | Logn | 1.7 |
| Oil production (comb) | CORINE_2012_Industrial_area | Logn | 1.7 |
| | TNO_PS for Oil production | Logn | 1.7 |
| Gas exploration (comb) | CORINE_2012_Industrial_area | Logn | 1.7 |
| | TNO_PS for Gas production | Logn | 1.7 |
| Coke ovens (comb) | CORINE_2012_Industrial_area | Logn | 1.7 |
| | TNO_PS for Iron and steel - Coke ovens | Logn | 1.7 |
| Non-ferrous metals (comb); Other non-ferrous metal production | CORINE_2012_Industrial_area | Logn | 3.7 |
| | TNO_PS for Non-ferrous metals - Other | Logn | 1.7 |
| Aluminium production | CORINE_2012_Industrial_area | Logn | 3.7 |
| | TNO_PS for Non-ferrous metals - Aluminium | Logn | 1.7 |
| Chemical industry (comb) | CORINE_2012_Industrial_area | Logn | 2.2 |
| | TNO_PS for Chemical industry | Logn | 1.7 |
| Passenger cars | RoadTransport_PassengerCars | Norm | 0.3 |
| Light duty vehicles | RoadTransport_LightCommercialVehicles | Norm | 0.3 |
| Trucks (>3.5t) | RoadTransport_HeavyDutyTrucks | Norm | 0.3 |



| | | | |
|---|---|---|---|
| Buses | RoadTransport_Buses | Norm | 0.3 |
| Motorcycles | RoadTransport_Motorcycles | Norm | 0.3 |
| Mopeds | RoadTransport_Mopeds | Norm | 0.5 |
| Civil aviation – LTO | Airport distribution for year 2015 | Logn | 1.4 |
| Mobile sources in agriculture/forestry/fishing | CORINE_2012_Arable_land | Logn | 1.4 |
| Other transportation, including pipeline compressors | Population_total_2015 | Logn | 3.7 |
| Small combustion - Residential - Household and gardening; Other mobile combustion | Population_total_2015 | Logn | 1.3 |
| Commercial/institutional | Population_total_2015 | Norm | 0.5 |
| | Population_rural_2015 | Logn | 1.3 |
| | Population_urban_2015 | Logn | 1.3 |
| | Wood_use_2014 | Logn | 2.2 |
| Residential | Population_total_2015 | Norm | 0.5 |
| | Population_rural_2015 | Logn | 1.3 |
| | Population_urban_2015 | Logn | 1.3 |
| | Wood_use_2014 | Logn | 1.4 |
| Agriculture/Forestry/Fishing | CORINE_2012_Arable_land | Logn | 1.4 |
| | Wood_use_2014 | Logn | 2.2 |
| Other stationary combustion | Population_total_2015 | Logn | 1.3 |
| | Population_rural_2015 | Logn | 1.3 |
| | Wood_use_2014 | Logn | 1.4 |
| Field burning of agricultural residues | CORINE_2012_Arable_land | Logn | 2.2 |
| | Population_total_2015 | Logn | 2.2 |
| Open burning of waste | CORINE_2012_Industrial_area | Logn | 3.7 |
| | Population_rural_2015 | Logn | 3.7 |

**Author contribution**

A.J.H. Visschedijk assembled the uncertainty data used in this work. S.N.C. Dellaert and H.A.C. Denier van der Gon are responsible for the base emission inventory. I. Super designed the experiments, carried them out, and prepared the manuscript with contributions from all co-authors.

**Competing interests**

The authors declare that they have no conflict of interest.

**Acknowledgements**

This study was supported by the $CO_2$ Human Emissions (CHE) project, funded by the European Union's Horizon 2020 research and innovation programme under grant agreement No 776186 and the VERIFY project, funded by the European Union's Horizon 2020 research and innovation programme under grant agreement No 776810.

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
