# Peer review of "Uncertainty analysis of a European high-resolution emission"

_Atmospheric Chemistry and Physics, 2019_

## Referee Comment (RC1) · Anonymous Referee #1 · 18 Oct 2019

Review of "Uncertainty analysis of a European high-resolution emission inventory of CO2 and CO to support inverse modelling and network design" by Super et al.

This manuscript describes an effort to construct an anthropogenic CO2 and CO inventory for a portion of Europe with carefully constructed uncertainties. The authors also show some basic analysis of their results, comparing uncertainties in different sectors and between countries, and the effect of some uncertainties on concentrations on CO2 or CO in the atmosphere. It is well-written, relevant, and extremely thorough, and should be published in ACP. The only major comment I have is about the data availability statement. The data availability requirement for publication has not been met:

[Figure]

data is only available by request to authors, which is not acceptable to this journal, I believe. Even if it is, I think the data (i.e. the inventory and uncertainties) should be made available publicly and without restriction, especially as I think this product would be of interest to many researchers.

Otherwise, my comments are fairly minor, and detailed below.

Introduction:

Please define TNO the first time to define the acronym for international readers.

L33 - How are the national numbers determined for reporting? These are also inventories, presumably of the scaled-up variety? perhaps the authors can make this section more specific to inventories that are spatially gridded and temporally downscaled, perhaps those commonly used for atmospheric studies?

L51: I am left wondering what a Tier 3 consists of in this regard, which the US EPA follows I believe.

L72: What is H2020?

L72: Should be made public, not on request - Journal editors can decide on this but that is my understanding of current publishing policy.

L70-76: These sentences are not actually very clear as to what the work is and confuse the reader. Are the 10 inventories part of this work, or only the new high-resolution inventory for the zoom region? No doubt this will be made clear later in the paper but should be outlined here.

L81: Should read: ... (12-16h LT) emissions, which could be the only emissions optimized in a study with a small domain, such as a city, using only afternoon observations? [if a study is regional or the city is large, then using mid-afternoon observations will still allow optimization of early morning emissions for example, depending on wind speed and location of emissions relative to the measurement point, for example]. But

I absolutely agree that looking at the temporal variability and whether that is correct can be crucial in an urban study as well as a regional one (as illustrated by Hu et al. Science Advances 2019 for continental work). It may be an issue even if the inversion is sensitive to all hours.

L108. Comma should be a period.

L108: if it's not described later, an additional sentence on the temporal disaggregation would be nice (does it account for weekday/weekend effects for example?).

L109: What is GNFR vs. NFR?

L138: is the point source data also temporally explicit? I am specifically thinking of energy generation (e.g. gas or coal-fired power plants, whose hour-to-hour emissions can vary drastically with no predictable cycle, at least in the U.S.).

Fig. 2: I understand from the text that correlations between sub-sectors are accounted for, but as this (and the next) figure shows aggregated sectors and no off-diagonal terms (i.e. no correlations in the uncertainties between sectors), why show these in this manner? Is the color axis in units of emissions, or do they range from 0-1 because they are covariances? (I would think the former, or they would all be 1 on the diagonal?). Or am I missing something here. Please clarify.

L218: should read "it is important to ensure"...

L234 define MC as Monte Carlo earlier

Fig 7 & 8: Captions should indicate left and right panels, for example "Contribution of source sectors to the total uncertainty in CO2 (left) and CO (right) emissions, summing to 100 %."

Same for Fig 9, it is easy enough to just say (left) and (right) in the caption here.

To clarify for fig 10 & 11, these spreads in concentration are from the experiments using different random emissions maps, i.e. the model was run 500 times, correct?

[Figure]

Fig 11 - I find this to be a very interesting analysis. It points to whether we expect an inversion to identify the true location of these 20 plants among all these scenarios. I.e. can some of the maps be shown to be false by the observed CO2? The large spread indicates maybe so, but then again, once all the other sources and their uncertainty are included, it would likely be pretty hard!

Section 3.3: inversion usually does not only include time from 12-16, just because those are the observation times. You may be optimizing emissions from earlier in the day, depending on the domain size and wind speeds. This should just be noted.

Fig 12 and text related: this standard time profile (black) seems to have a monthly mean that is then also distributed hourly through the day? Is it hourly, or 3-hourly? weekday/weekend (for businesses vs. residences, assuming those are contributing to stationary combustion for on-site heating, e.g. burning of gas)? These details could be mentioned in the caption, I realize they are not necessary for describing the uncertainty method, but they are useful to know for users of the inventory emissions.

Fig 14, the text on the map, especially "Tirol" and "Hotspot", is hard to see here - perhaps larger, or placed in a different section with an arrow to the appropriate box?

L475: Does this statement refer to their methods for calculating emissions or uncertainties?

Data Availability: See note at top, this data should be publicly available on a public-facing data portal.

---

## Referee Comment (RC2) · Anonymous Referee #2 · 21 Oct 2019

This manuscript presents an assessment of the uncertainties in high-resolution emission inventories of CO2 and CO in 14 European countries. The uncertainties present in various underlying parameters of the inventories (e.g., absolute uncertainties in reported emissions, emission factors, spatial proxies, temporal profiles) are propagated using a well-described Monte Carlo simulation routine. The uncertainties are tracked to assess the importance of specific source sectors in introducing large uncertainties on both absolute and relative bases. Several factors are found to be playing an important role in contributing to the final emission uncertainties. For instance, spatial disaggregation of the emissions at a high spatial resolution results in large uncertainties at the local/city scale, which has important implications for inverse modeling studies operat-

ing within these smaller spatial domains. The authors find that because certain sectors with large overall contribution of $CO_2$ and CO emissions are well-constrained (e.g., industrial sector in the Paris metro area), the relative uncertainties in these locations are far smaller than those in the immediate rural surroundings. Thus, future efforts to reduce absolute emissions of $CO_2$ and CO may use the absolute uncertainties presented in this study to identify a network of key target areas/sectors.

Overall, I find this manuscript well-written. The methods are presented in sufficient detail that one could reproduce them. The methods section lacks some quality assurance, as I have described in my first major comment. The results are presented in an organized fashion, and the interpretations and conclusions are generally well-reasoned. My second major comment has to do with the framing of these interpretations with respect to observation-based literature, especially since the authors mention a motivation for this study is to facilitate inter-comparison of modeled and observed greenhouse gas concentrations. Once the authors have addressed these comments adequately, the manuscript should be ready for publication in ACP. In addition to my two major comments, I list several minor comments that are mostly typographical errors and/or suggestions to improve presentation of figures and tables.

**Major comments:**

1. A rationale for number of Monte Carlo (MC) simulations should be provided. For e.g., a plot with some metric of quality of results (total residual, total error, full-width half maximums of the distributions showed in Fig. 7, etc.) versus number of MC runs. I'd expect such a plot to have an exponential decay with respect to increasing MC runs, which would then help justify the choice of N = 500.

2. I think the presentation of the results could be better framed with respect to other literature. For instance, the authors show that the spread in their modeled $CO_2$ and CO concentrations reduces over distances of 5–40 km from the source categories (Figures 10 and 11). How does this length-scale compare with other

studies? Are there any monitoring studies that have shown similar fall-off length-scales? To say that road transport affects CO concentration as far as 40 km downwind seems excessive, if one were to compare it to, say, Figure 4A from Canagaratna et al.'s mobile monitoring study.

**Minor comments:**

1. L24: I suggest using "abundant", instead of "important". In terms of warming potential, there are other gases more important than CO2 (e.g., CH4).

2. L36: "report", not "reported".

3. L69: "atmospheric", not "atmospherics".

4. L64-65: "in contrast, if . . . are needed." This sentence is unclear. What is "prior" referring to? Please reword.

5. L76-77: not sure what "European zoom region" means. Please clarify.

6. L81: suggest replacing "time profiles" with "temporal profiles", or "diurnal profiles".

7. L81-83: question 3 is somewhat unclear. It could be reworded for clarity, but also the motivation for this question was not set up in the introduction. This makes this question feel abruptly added.

8. L86: is "partitioning" the right word? I suggest using "apportionment", instead.

9. Table 1: for consistency with first usage in L26, please continue with the "FFCO", "FFCO2" naming conventions. The acronyms FF and BF should be declared in the Table caption. Also, is there a specific reason to use three-letter country codes, instead of simply country names?

10. L103: "gap-filled" (should be hyphenated).

11. L103: suggest replacing "data was gap filled" with something more informative of which attributes of the dataset were missing, and how they were filled (i.e., with NaNs, or geospatial interpolated, etc.).

12. L105: acronym "AIS" not defined.

13. L108: comma is used instead of period.

14. L109: acronym "GNFR" not defined, and is also used in Table 2 without definition.

15. Figure 1: could the authors add a few landmarks or identify a few of the visible hotspots in these maps e.g., Paris? It'd be helpful for a reader not familiar with the placement of major urban areas of Europe.

16. L148: which "differences" are being referred to in this sentence? Differences in uncertainties, I assume? Should be clarified.

17. Figures 2 and 3: the gridlines need to match the category labels. The current version of this figure is difficult to read easily.

18. L294-296: This sentence is confusing. Here is how I would calculate the "uncertainty in the total emissions": a) take the standard deviations of emissions from each sector (i.e., standard deviation of each box in Figure 7-left), b) calculate the average of the standard deviations from (a), and c) report this average from (b) as "uncertainty in total emissions". However, it seems the authors have used a different approach: a) take the standard deviations of emissions from each sector (i.e., standard deviation of each box in Figure 7-left), b) calculate the STANDARD DEVIATION of the standard deviations from (a), and c) report this STANDARD DEVIATION from (b) as "uncertainty in total emissions". Is this correct? If yes, this would not be the "uncertainty in total emissions", but rather would be an "uncertainty of the uncertainty in total emissions". Please justify/clarify/correct in the manuscript accordingly.

19. Figures 8, 9, and 16: As I indicated in my initial review prior to posting on ACPD, the legend needs to be reversed to be consistent with the order of stacking.

20. L322-323: "Overall, the differences between countries are relatively small (Figure 9, right panel)." Instead of using a qualitative term like "relatively small", why not report the total uncertainty as done for Figure 7?

21. L339: "For CO2 (left panel) we see a concentration of about . . ." should be changed to "For CO2 (left panel), we see a spread in concentration of about . . ." There is no information in Figure 10 about the absolute CO2 concentrations, so seeing a concentration of 3 ppm anywhere in ambient air would be impossible. A related suggestion is to plot the absolute numbers on the right axis, to get a better sense of the absolute concentrations (especially for CO).

22. L343: what does "atmospheric signal" mean? Why not just say "modeled spread in concentration"?

23. Figures 10 and 11: the x-axis labels are unevenly spaced. It is definitely not linear, but it doesn't seem log-spaced either. Please correct/clarify in Figure caption.

24. Figure 12: Please describe what the grey lines represent in the caption.

25. L397: "in inverse modeling, often . . . transport". It'd be good to include a couple of references to support this statement.

26. Figure 13: I know that time profiles used for modeling CO2 and CO emissions are the same, but it'd be good to rename the y-axis label to "normalized spread in CO2 and CO emissions", and remind the reader of this very briefly in the caption.

27. L447: "big cities like Paris, Berlin, and Brussels". I assume this is about the CO, and not the CO2 map? It'd help to point to these cities in Figure 15.

28. L528: no need to define the acronym LTO, if it is only used once.

**Reference(s):**

Canagaratna, M. R.; Onasch, T. B.; Wood, E. C.; Herndon, S. C.; Jayne, J. T.; Cross, E. S.; Miake-Lye, R. C.; Kolb, C. E.; Worsnop, D. R. Evolution of vehicle exhaust particles in the atmosphere. J. Air Waste Manag. Assoc. 2010, 60 (10), 1192–1203.

---

## Author Comment (AC1) · 14 Jan 2020

We would like to thank the reviewers for their enthusiasm about our study and for the comments on our work. The review comments have been helpful in reflecting on our work and pointing out parts that required further improvements. Below we address specific issues mentioned by the reviewers point by point. The manuscript has been updated accordingly (changes are highlighted).

***Anonymous Referee #1

*Review of "Uncertainty analysis of a European high-resolution emission inventory of CO2 and CO to support inverse modelling and network design" by Super et al.*

*This manuscript describes an effort to construct an anthropogenic CO2 and CO inventory for a portion of Europe with carefully constructed uncertainties. The authors also show some basic analysis of their results, comparing uncertainties in different sectors and between countries, and the effect of some uncertainties on concentrations on CO2 or CO in the atmosphere. It is well-written, relevant, and extremely thorough, and should be published in ACP. The only major comment I have is about the data availability statement. The data availability requirement for publication has not been met: data is only available by request to authors, which is not acceptable to this journal, I believe. Even if it is, I think the data (i.e. the inventory and uncertainties) should be made available publicly and without restriction, especially as I think this product would be of interest to many researchers.*

We thank the reviewer for this suggestion. We agree that the data is useful for many researchers and have made the data accessible through Zenodo (see Data Availability description).

*Otherwise, my comments are fairly minor, and detailed below.*

*Introduction: Please define TNO the first time to define the acronym for international readers.*

We have replaced TNO by 'the Netherlands Organisation for Applied Scientific Research (TNO)' in lines 35-36.

*L33 - How are the national numbers determined for reporting? These are also inventories, presumably of the scaled-up variety? perhaps the authors can make this section more specific to inventories that are spatially gridded and temporally downscaled, perhaps those commonly used for atmospheric studies?*

The reported country-level emissions are available from UNFCCC ([https://unfccc.int/process-and-meetings/transparency-and-reporting/reporting-and-review-under-the-convention/greenhouse-gas-inventories-annex-i-parties/national-inventory-submissions-2019](https://unfccc.int/process-and-meetings/transparency-and-reporting/reporting-and-review-under-the-convention/greenhouse-gas-inventories-annex-i-parties/national-inventory-submissions-2019)) and based on energy statistics and emission factors following IPCC guidelines and are calculated as national, yearly total. This has been clarified in lines 32-34. In some cases IPCC default values are used, but countries can also decide to use country-specific values (which are generally more realistic). There is no scaling involved here, except that the overall energy consumption for one sector is divided over several sub-sectors. This introduces some uncertainty, which is taken into account in the uncertainty definition for the activity data listed in Appendix A.

*L51: I am left wondering what a Tier 3 consists of in this regard, which the US EPA follows I believe.*

It should be noted that there are two types of calculations with respect to emission reporting: calculation of the emissions and calculation of the emission uncertainties. The IPCC describes only the Tier 1 and Tier 2 approach for calculating emission uncertainties. In contrast, there is also a Tier 3 approach for calculation of emissions, which uses country-specific data and models. The Tier 3 approach used by the US EPA is therefore related to calculation of the emissions, and not of the emission uncertainties. In short, there are 3 Tiers for emission calculation and 2 Tiers for uncertainties.

*L72: What is H2020?*

H2020 is short for Horizon 2020, a European Research and Innovation programme. This is clarified in lines 77-78.

*L72: Should be made public, not on request - Journal editors can decide on this but that is my understanding of current publishing policy.*

We agree and have made the data accessible through Zenodo (see Data Availability description).

*L70-76: These sentences are not actually very clear as to what the work is and confuse the reader. Are the 10 inventories part of this work, or only the new high-resolution inventory for the zoom region? No doubt this will be made clear later in the paper but should be outlined here.*

We have clarified that the methodology used to create the family of emission inventories is also used in the work described in this manuscript (lines 78-79).

*L81: Should read: ... (12-16h LT) emissions, which could be the only emissions optimized in a study with a small domain, such as a city, using only afternoon observations? [if a study is regional or the city is large, then using mid-afternoon observations will still allow optimization of early morning emissions for example, depending on wind speed and location of emissions relative to the measurement point, for example]. But I absolutely agree that looking at the temporal variability and whether that is correct can be crucial in an urban study as well as a regional one (as illustrated by Hu et al. Science Advances 2019 for continental work). It may be an issue even if the inversion is sensitive to all hours.*

For clarity this topic has been introduced earlier in the introduction (lines 66-70), so that more explanation can be given.

*L108. Comma should be a period.*

Done.

*L108: if it's not described later, an additional sentence on the temporal disaggregation would be nice (does it account for weekday/weekend effects for example?).*

The time profiles are described in more detail in Section 2.2.4 (lines 207-213). Although a weekly cycle is included, e.g. with lower traffic emissions during the weekend, the diurnal profile is the same for weekdays and weekends.

*L109: What is GNFR vs. NFR?*

NFR sectors are very detailed and aggregated to GNFR sectors, i.e. GNFR is the aggregated version of NFR used for delivering gridded inventory data (the "G" stands for Gridding) (line 119). This is now also mentioned in the caption of Table 2.

*L138: is the point source data also temporally explicit? I am specifically thinking of energy generation (e.g. gas or coal-fired power plants, whose hour-to-hour emissions can vary drastically with no predictable cycle, at least in the U.S.).*

The point sources get the same temporal distribution as the area sources, using the fixed time profiles. This has been made more clear in lines 148-149. We have previously studied daily activity from some major power plants and found that, indeed, the temporal variations in emissions from power plants is difficult to describe with environmental variables. Unfortunately, temporally detailed activity data is not broadly available and therefore not part of the emission inventory. This is an important point for the future, which is now also mentioned in lines 594-595.

*Fig. 2: I understand from the text that correlations between sub-sectors are accounted for, but as this (and the next) figure shows aggregated sectors and no off-diagonal terms (i.e. no correlations in the uncertainties between sectors), why show these in this manner? Is the color axis in units of emissions, or do they range from 0-1 because they are covariances? (I would think the former, or they would all be 1 on the diagonal?). Or am I missing something here. Please clarify.*

Error correlations exist between several sub-sectors that are part of the same aggregated (GNFR) sector. Therefore, no off-diagonal values are visible, i.e. there are no error correlations between GNFR sectors. The reason that the uncertainties are displayed in this manner (covariance matrix) is because this is the common way to describe prior uncertainties for inversion studies. They are indeed covariances, ranging from 0 to 1.

*L218: should read "it is important to ensure"...*

Done.

*L234 define MC as Monte Carlo earlier*

The abbreviation MC has been replaced with the full name 'Monte Carlo' throughout the manuscript.

*Fig 7 & 8: Captions should indicate left and right panels, for example "Contribution of source sectors to the total uncertainty in CO2 (left) and CO (right) emissions, summing to 100 %."*

*Same for Fig 9, it is easy enough to just say (left) and (right) in the caption here.*

These indications have been added consistently to all figure captions.

*To clarify for fig 10&11, these spreads in concentration are from the experiments using different random emissions maps, i.e. the model was run 500 times, correct?*

Yes, this is correct. This is now mentioned in line 293-294 and the figure captions.

*Fig 11 - I find this to be a very interesting analysis. It points to whether we expect an inversion to identify the true location of these 20 plants among all these scenarios. I.e. can some of the maps be shown to be false by the observed CO2? The large spread indicates maybe so, but then again, once all the other sources and their uncertainty are included, it would likely be pretty hard!*

Yes, we agree that it will be very challenging. Also given the uncertainty introduced by the model transport, which would likely be similar or larger than (depending on the complexity of the area) the spread caused by the different maps. Nevertheless, we show here that it is an important source of uncertainty for local inversions.

*Section 3.3: inversion usually does not only include time from 12-16, just because those are the observation times. You may be optimizing emissions from earlier in the day, depending on the domain size and wind speeds. This should just be noted.*

We have added a sentence that explains that our reasoning only applies to local studies (lines 420-421).

*Fig 12 and text related: this standard time profile (black) seems to have a monthly mean that is then also distributed hourly through the day? Is it hourly, or 3-hourly? weekday/weekend (for businesses vs. residences, assuming those are contributing to stationary combustion for on-site heating, e.g. burning of gas)? These details could be mentioned in the caption, I realize they are not necessary for describing the uncertainty method, but they are useful to know for users of the inventory emissions.*

For each sector there are three temporal profiles: a profile describing the seasonal cycle (monthly factors), a profile describing day-to-day variations (daily factors) and a profile describing the diurnal cycle (hourly factors). This is now described in more detail in lines 207-213.

*Fig 14, the text on the map, especially "Tirol" and "Hotspot", is hard to see here perhaps larger, or placed in a different section with an arrow to the appropriate box?*

Both Fig. 14 and Fig. 15 have been improved by increasing the weight of the boxes and the font size.

*L475: Does this statement refer to their methods for calculating emissions or uncertainties?*

Actually both, but in this case we mean the uncertainty calculation. This is clarified in line 503.

*Data Availability: See note at top, this data should be publicly available on a public-facing data portal.*

Data are now publicly available. The Data Availability section has been updated accordingly.

*Anonymous Referee #2

*This manuscript presents an assessment of the uncertainties in high-resolution emission inventories of CO2 and CO in 14 European countries. The uncertainties present in various underlying parameters of the inventories (e.g., absolute uncertainties in reported emissions, emission factors, spatial proxies, temporal profiles) are propagated using a well-described Monte Carlo simulation routine. The uncertainties are tracked to assess the importance of specific source sectors in introducing large uncertainties on both absolute and relative bases. Several factors are found to be playing an important role in contributing to the final emission uncertainties. For instance, spatial disaggregation of the emissions at a high spatial resolution results in large uncertainties at the local/city scale, which has important implications for inverse modeling studies operating within these smaller spatial domains. The authors find that because certain sectors with large overall contribution of CO2 and CO emissions are well-constrained (e.g., industrial sector in the Paris metro area), the relative uncertainties in these locations are far smaller than those in the immediate rural surroundings. Thus, future efforts to reduce absolute emissions of CO2 and CO may use the absolute uncertainties presented in this study to identify a network of key target areas/sectors.*

*Overall, I find this manuscript well-written. The methods are presented in sufficient detail that one could reproduce them. The methods section lacks some quality assurance, as I have described in my first major comment. The results are presented in an organized fashion, and the interpretations and conclusions are generally well-reasoned. My second major comment has to do with the framing of these interpretations with respect to observation-based literature, especially since the authors mention a motivation for this study is to facilitate inter-comparison of modeled and observed greenhouse gas concentrations. Once the authors have addressed these comments adequately, the manuscript should be ready for publication in ACP. In addition to my two major comments, I list several minor comments that are mostly typographical errors and/or suggestions to improve presentation of figures and tables.*

*Major comments:*

*1. A rationale for number of Monte Carlo (MC) simulations should be provided. For e.g., a plot with some metric of quality of results (total residual, total error, fullwidth half maximums of the distributions showed in Fig. 7, etc.) versus number of MC runs. I'd expect such a plot to have an exponential decay with respect to increasing MC runs, which would then help justify the choice of N = 500.*

Because the Monte Carlo simulations performed in this study are relatively cheap the sample size can be taken large enough to ensure a robust result. However, to support our choice for N=500 we used two methods described in literature: sampling statistics and bootstrapping. Both methods give similar curves, such as shown below. The curve indicates the spread in the standard deviations if we would repeat the Monte Carlo simulation multiple times for a specific sample size, i.e. it indicates how robust the uncertainty estimate is. From this figure we conclude that a sample size larger than 500 would not increase the robustness a lot.

[Figure]

We have added a statement that a sample size of N=500 is large enough to get robust results, based on the analysis shown here (lines 249-250). For completeness, we have added this figure to Appendix B.

Reference:

Janssen, H.: Monte-Carlo based uncertainty analysis: Sampling efficiency and sampling convergence. Reliability Engineering & System Safety, 109, 123-132, https://doi.org/10.1016/j.ress.2012.08.003, 2013.

*2. I think the presentation of the results could be better framed with respect to other literature. For instance, the authors show that the spread in their modeled CO2 and CO concentrations reduces over distances of 5–40 km from the source categories (Figures 10 and 11). How does this length-scale compare with other studies? Are there any monitoring studies that have shown similar fall-off length-scales? To say that road transport affects CO concentration as far as 40 km downwind seems excessive, if one were to compare it to, say, Figure 4A from Canagaratna et al.'s mobile monitoring study.*

We agree with the reviewer that we have not discussed this topic in our paper. Instead, we have focused our discussion on the estimated uncertainties in the (gridded) emissions, which is the main topic of our paper and covers most of our results. However, we have put some effort in improving the comparison of the model results with other literature, mainly to explain how these model results can be used for network design and inverse modelling (lines 499-501 and lines 570-585).

*Minor comments:*

*1. L24: I suggest using "abundant", instead of "important". In terms of warming potential, there are other gases more important than CO2 (e.g., CH4).*

We have replaced this word.

*2. L36: "report", not "reported".*

The sentence has been rephrased for clarification (line 37).

*3. L69: "atmospheric", not "atmospherics".*

Done.

*4. L64-65: "in contrast, if ... are needed." This sentence is unclear. What is "prior" referring to? Please reword.*

In inverse modelling the word 'prior' refers to the initial emission inventory, containing information on the order of magnitude and location of emissions which is then updated using atmospheric observations. This has been clarified in line 65.

*5. L76-77: not sure what "European zoom region" means. Please clarify.*

We have clarified which region is meant in line 84.

*6. L81: suggest replacing "time profiles" with "temporal profiles", or "diurnal profiles".*

We have replaced 'time profiles' with 'temporal profiles' throughout the manuscript.

*7. L81-83: question 3 is somewhat unclear. It could be reworded for clarity, but also the motivation for this question was not set up in the introduction. This makes this question feel abruptly added.*

This topic has now been introduced in lines 66-70 for clarification.

*8. L86: is "partitioning" the right word? I suggest using "apportionment", instead.*

This word has been replaced in line 94.

*9. Table 1: for consistency with first usage in L26, please continue with the "FFCO", "FFCO2" naming conventions. The acronyms FF and BF should be declared in the Table caption. Also, is there a specific reason to use three-letter country codes, instead of simply country names?*

The names of the species have been updated and FF and BF are explained in the caption of Table 1. Also, the names of the countries have been given.

*10. L103: "gap-filled" (should be hyphenated).*

Done.

*11. L103: suggest replacing "data was gap filled" with something more informative of which attributes of the dataset were missing, and how they were filled (i.e., with NaNs, or geospatial interpolated, etc.).*

The gap-filling refers to the reported emissions, which are sometimes missing for specific sectors. Also, in some cases the reported data is considered to be unreliable. In those cases, other sources of information are used to update and complete the emission inventory. This is now explained in more detail in lines 112-113.

*12. L105: acronym "AIS" not defined.*

The acronym has been defined in line 115.

*13. L108: comma is used instead of period.*

The comma has been replaced.

*14. L109: acronym "GNFR" not defined, and is also used in Table 2 without definition.*

GNFR is the aggregated version of NFR. This is now mentioned in the caption of Table 2.

*15. Figure 1: could the authors add a few landmarks or identify a few of the visible hotspots in these maps e.g., Paris? It'd be helpful for a reader not familiar with the placement of major urban areas of Europe.*

Figure 1 has been updated and indicates the location of some major urban areas (Paris, Ruhr area and Rotterdam).

*16. L148: which "differences" are being referred to in this sentence? Differences in uncertainties, I assume? Should be clarified.*

Indeed, this has been made explicit in line 159-160.

*17. Figures 2 and 3: the gridlines need to match the category labels. The current version of this figure is difficult to read easily.*

We choose to match the gridlines with the category labels, such that the intersection of two lines clearly indicates the value belonging to those categories. But if we understand correctly the reviewer would like to see the gridlines surrounding each entry in the matrix, instead of being in the middle of each row/column. This has been updated.

*18. L294-296: This sentence is confusing. Here is how I would calculate the "uncertainty in the total emissions": a) take the standard deviations of emissions from each sector (i.e., standard deviation of each box in Figure 7-left), b) calculate the average of the standard deviations from (a), and c) report this average from (b) as "uncertainty in total emissions". However, it seems the authors have used a different approach: a) take the standard deviations of emissions from each sector (i.e., standard deviation of each box in Figure 7-left), b) calculate the STANDARD DEVIATION of the standard deviations from (a), and c) report this STANDARD DEVIATION from (b) as "uncertainty in total emissions". Is this correct? If yes, this would not be the "uncertainty in total emissions", but rather would be an "uncertainty of the uncertainty in total emissions". Please justify/clarify/correct in the manuscript accordingly.*

The uncertainty in the total emissions is calculated from the Monte Carlo, similarly as the uncertainty for each sector. The Monte Carlo provides 500 solutions for each of the defined sub-sectors (NFR – fuel combination). These are summed to get 500 solutions for GNFR sector emissions. If we then sum the GNFR sector emissions we get 500 solutions for the total emission. What is reported here is the standard deviation of these 500 solutions. We have clarified this in lines 309-311.

*19. Figures 8, 9, and 16: As I indicated in my initial review prior to posting on ACPD, the legend needs to be reversed to be consistent with the order of stacking.*

The order of the legends has been reversed.

*20. L322-323: "Overall, the differences between countries are relatively small(Figure 9, right panel)." Instead of using a qualitative term like "relatively small", why not report the total uncertainty as done for Figure 7?*

We have presented the range of standard deviations, which is between 1.2 and 2.3% (lines 339-341).

*21. L339: "For CO2 (left panel) we see a concentration of about ..." should be changed to "For CO2 (left panel), we see a spread in concentration of about ..." There is no information in Figure 10 about the absolute CO2 concentrations, so seeing a concentration of 3 ppm anywhere in ambient air would be impossible. A related suggestion is to plot the absolute numbers on the right axis, to get a better sense of the absolute concentrations (especially for CO).*

We agree that these concentrations are not what you would measure in the atmosphere. Rather, they are the result only of the emissions of stationary combustion (CO2) and road transport (CO). This has been clarified in the text (lines 357-358).

*22. L343: what does "atmospheric signal" mean? Why not just say "modeled spread in concentration"?*

What we refer to is the concentration enhancement caused by the specified sector. We have replaced 'signal' by 'concentration enhancement' throughout the text.

*23. Figures 10 and 11: the x-axis labels are unevenly spaced. It is definitely not linear, but it doesn't seem log-spaced either. Please correct/clarify in Figure caption.*

The labels are unevenly spaced, because we only added labels for the distances that we have included in the model simulations. Nevertheless, the axis is linear.

*24. Figure 12: Please describe what the grey lines represent in the caption.*

The grey lines represent the spread in the temporal profile, resulting from the Monte Carlo simulation. This is clarified in the caption.

*25. L397: "in inverse modeling, often ... transport". It'd be good to include a couple of references to support this statement.*

See lines 419-420 for the references.

*26. Figure 13: I know that time profiles used for modeling CO2 and CO emissions are the same, but it'd be good to rename the y-axis label to "normalized spread in CO2 and CO emissions", and remind the reader of this very briefly in the caption.*

We have added this comment to the figure caption.

*27. L447: "big cities like Paris, Berlin, and Brussels". I assume this is about the CO, and not the CO2 map? It'd help to point to these cities in Figure 15.*

Indeed, this part describes relative uncertainties in CO (as is also mentioned in the caption of Figure 17). We have added this to line 472. For clarity, we have replaced the current list with those urbanized areas now shown in Fig. 1, including a reference to this figure (line 471-472).

*28. L528: no need to define the acronym LTO, if it is only used once.*

The term LTO has been removed.

*Reference(s): Canagaratna, M. R.; Onasch, T. B.; Wood, E. C.; Herndon, S. C.; Jayne, J. T.; Cross, E. S.; Miake-Lye, R. C.; Kolb, C. E.; Worsnop, D. R. Evolution of vehicle exhaust particles in the atmosphere. J. Air Waste Manag. Assoc. 2010, 60 (10), 1192–1203.*